# An on-demand bioresorbable neurostimulator

**Dong-Min Lee** [1,2,10], **Minki Kang** [3,10], **Inah Hyun** [1,2,10], **Byung-Joon Park**[1,2], **Hye Jin Kim**[4], **Soo Hyun Nam**[5], **Hong-Joon Yoon** [6], **Hanjun Ryu** [7], **Hyun-moon Park**[8], **Byung-Ok Choi**[4,5,9] ✉ **& Sang-Woo Kim** [1,2] ✉

Bioresorbable bioelectronics, with their natural degradation properties, hold significant potential to eliminate the need for surgical removal. Despite notable achievements, two major challenges hinder their practical application in medical settings. First, they necessitate sustainable energy solutions with biodegradable components via biosafe powering mechanisms. More importantly, reliability in their function is undermined by unpredictable device lifetimes due to the complex polymer degradation kinetics. Here, we propose an on-demand bioresorbable neurostimulator to address these issues, thus allowing for clinical operations to be manipulated using biosafe ultrasound sources. Our ultrasound-mediated transient mechanism enables (1) electrical stimulation through transcutaneous ultrasound-driven triboelectricity and (2) rapid device elimination using high-intensity ultrasound without adverse health effects. Furthermore, we perform neurophysiological analyses to show that our neurostimulator provides therapeutic benefits for both compression peripheral nerve injury and hereditary peripheral neuropathy. We anticipate that the on-demand bioresorbable neurostimulator will prove useful in the development of medical implants to treat peripheral neuropathy.

Electric signals play a major role in the nervous system, facilitating vital functions throughout the human body: controlling organ motility, maintaining homeostasis, and accepting sensations[1]. The use of electric signals for therapeutic purposes has been widely adopted to modulate the physiological mechanism of the nervous system[2]. Therefore, electroceuticals, a branch of devices that deliver electric signals to the targeted neural tissue, have been developed as an alternative pharmacological treatment[3,4]. In particular, the miniaturized implantable devices that employ bioresorbable materials, termed bioresorbable electroceuticals, have gained significant interest, as they meet the clinical need for short-term treatment to treat neuropathy based on minimized adverse health effects and avoid device extraction surgery[5–9]. For example, previous studies suggested that radiofrequency (RF)-powered bioresorbable devices enabled effective nonpharmacological electrotherapy to promote the recovery of the peripheral nerve from injuries[10,11]. In addition, an electrically assisted bioresorbable microfluidic device established a targeted and reversible pain management system[12]. However, the practical application of conventional bioresorbable electronics for clinical purposes has been hindered by the inherent challenge of predicting device lifespan,

[1]Department of Materials Science and Engineering, Yonsei University, Seoul 03722, Republic of Korea. [2]Center for Human-oriented Triboelectric Energy Harvesting, Yonsei University, Seoul 03722, Republic of Korea. [3]School of Advanced Materials Science and Engineering, Sungkyunkwan University (SKKU), Suwon 16419, Republic of Korea. [4]Department of Neurology, Samsung Medical Center, Sungkyunkwan University School of Medicine, Seoul 06351, Republic of Korea. [5]Cell and Gene Therapy Institute (CGTI), Samsung Medical Center, Seoul 06351, Republic of Korea. [6]Department of Electronic Engineering, Gachon University, Seongnam 13120, Republic of Korea. [7]Department of Advanced Materials Engineering, Chung-Ang University, Anseong 17546, Republic of Korea. [8]Research and Development Center, Energy-Mining Co., LTD., Suwon 16226, Republic of Korea. [9]Samsung Advanced Institute for Health Sciences & Technology (SAIHST), Seoul 06351, Republic of Korea. [10]These authors contributed equally: Dong-Min Lee, Minki Kang, Inah Hyun. ✉e-mail: bochoi@skku.edu; kimsw1@yonsei.ac.kr

which arises from the diverse transient characteristics exhibited by their constituent materials.

To achieve next-step breakthroughs for bioresorbable electroceuticals, the realization of on-demand management is required to control the device lifespan through material design and well-defined triggering events. Conventional bioresorbable electroceuticals are absorbed in the body over time by passive operation, a system whose lifetime relies on the gradual degradation determined by the dimensions of constituent materials and their degrading properties. However, this passive operation system may endow an undesirable burden on patients, because the prolonged residues may induce inflammation or toxicity to human tissues[13]. In contrast, active operation, a system whose operation can be accurately controlled upon well-designed triggering events, facilitates the complete management of the device according to the programmed clinical timeline. Moreover, it has gained significant attention for its clinical safety, as potential harm can be minimized by eliminating prolonged residues[13].

Herein, we propose an on-demand bioresorbable neurostimulator that exhibits the following key advances: (1) a battery-free platform that exploits the triboelectric mechanism using medically available ultrasound sources, (2) pin-point electrotherapy on peripheral nerve with secured biological safety, and (3) an on-demand device elimination that enables non-invasive adjustment of clinical plans. Our device consists of two main parts, an acoustically triggerable transient triboelectric nanogenerator (ACT−TENG), and a bioresorbable cuff electrode (Fig. 1a). The ACT−TENG was developed to represent a bioresorbable energy harvesting system using a biocompatible ultrasound source for its non-invasive elimination[14,15]. The bioresorbable cuff electrode, composed of a pair of magnesium (Mg) electrodes and poly(3-hydroxybutyrate-co-3-hydroxyvalerate) (PHBV) membranes, delivers the electrical impulses from the ACT−TENG to the targeted site of the sciatic nerve (Supplementary Fig. 1). The ACT−TENG generates high-frequency triboelectric impulses (20 kHz, AC waveform), offering effective nonpharmacological therapy for peripheral neuropathies. We place a particular focus on its therapeutic effect in both compression peripheral nerve injury and hereditary peripheral neuropathy including Charcot−Marie−Tooth disease (CMT). CMT is one of the most common hereditary peripheral neuropathies and CMT type 1 A (CMT1A) is caused by duplication of the peripheral myelin protein (*PMP22*) gene[16]. CMT1A, which accounts for approximately 50% of CMT, has more than 1.4 million patients worldwide[17]. To date, CMT1A is known as an incurable disease and there is no FDA-approved treatment[18]. Although kHz frequency stimulation with sinusoidal waveform has been employed as a peripheral nerve stimulation (PNS) system by engaging ion channel dynamics of the targeted axons, its therapeutic effect on both compression nerve injury and hereditary peripheral neuropathy has remained undiscovered[19,20].

## Results

### Design and clinical application of the on-demand bioresorbable neurostimulator

We utilized PHBV membranes due to their low dissolution rate (about 10% weight loss in 450 days), allowing more practical manipulation of the device lifetime (see more descriptions about the transient mechanism of PHBV membranes in Supplementary Note 2)[21]. We achieved the high triboelectric performance of the ACT−TENG through the addition of polyethylene glycol (PEG) and choline chloride (ChCl) in PHBV (to form PHBV/PEG:ChCl composite) (Supplementary Fig. 2). On a molecular basis, the presence of hydroxyl functional groups (−OH) in PEG help improve the surface potential due to their electron-donating property[22]. Also, ChCl, an edible salt, contributes significantly to the high surface potential and high surface charge density (Supplementary Fig. 3)[23]. The underlying mechanism is based on the mobile chlorine anions induced by positive triboelectric charges at the surface, while choline cations are immobilized by forming hydrogen bonding with polymeric chain networks (Supplementary Fig. 4)[24,25]. However, the excessive content of ChCl (more than 0.05 m) caused mechanical instability, which interrupts consistent energy-generating performance (Supplementary Figs. 5 and 6). Figure 1b illustrates the triboelectric mechanism upon ultrasound-driven vibrating motions. The propagated ultrasonic wave induces the high-frequency contact/separation of the PHBV/PEG:ChCl triboelectric layer with the Mg electrode (Supplementary Figs. 7−9)[26,27]. Submerged into deionized water at 5 mm under an ultrasound probe, the ACT−TENG generated 7.8 V voltage peaks at 40 megohm impedance and 60.0 μA current peaks at 1 ohm impedance (Supplementary Movie 1 and Supplementary Figs. 10 and 11). We further confirmed that the ACT−TENG produces a power density of 102.4 μW cm$^{-2}$ at an optimal impedance of 100 kilohms.

Figure 1c displays a series of schematics of our proposed on-demand electrotherapy to treat peripheral neuropathies. During the clinical schedule, as a low-intensity ultrasonic wave (≤1.0 W cm$^{-2}$) propagates through the deep tissue from an external ultrasound probe to the implanted ACT−TENG, the resulting triboelectric signals flow through an Mg lead of the cuff electrode. After its clinical role, we intend to employ a high-intensity ultrasonic wave (HIU, ≥3.0 W cm$^{-2}$) to accelerate the mechanical disintegration of the neurostimulator. First, the HIU introduces a locally focused acoustic pressure inside the porous structure of the PHBV encapsulation layer, providing a favorable condition for the degradation event (Supplementary Figs. 12 and 13). Then, an enlarged surface area associated with the HIU-driven disintegration promotes the hydrolytic dissolution of the constituent materials (Fig. 1d, Supplementary Fig. 14). We experimentally demonstrated the distinct responsiveness of the on-demand bioresorbable neurostimulator in deionized water. Figure 1e shows that the ACT−TENG generates stable electrical output performances for 120 min at 0.5 W cm$^{-2}$, while it loses its energy-generating function within 20 min upon the HIU. Figure 1f depicts a summation plot of Fig. 1e, showing the on-demand use of electrotherapy according to the clinical schedule. In vitro demonstration of the HIU-triggered transience suggests that the device in diluted phosphate-buffered saline (PBS, pH 7.4) solution underwent rapid mechanical degradation and full mechanical disintegration in 120 min (Fig. 1g, Supplementary Fig. 15, and Supplementary Movie 2). In contrast, without any triggering events, it solely maintained its structure in diluted PBS solution at 75 °C for 20 days (d), due to the relatively low dissolution rate of PHBV (Supplementary Fig. 16).

### In vivo demonstrations of HIU-triggered transient mechanism and its biosafety

To verify whether the on-demand bioresorbable neurostimulator also works in a living organism, we investigated its in vivo transient processes and resulting immune responses. We took a series of micro-CT images of the device implanted in a mouse model after the HIU triggering events (Fig. 2a, Supplementary Fig. 17). We observed that the device mechanically disintegrated, ending up in complete elimination in 120 min. In hematoxylin and eosin (H&E)-stained tissues, both the passive operation group and the HIU-triggered group exhibited little acute inflammation as a similar degree of inflammatory cells (e.g., macrophages and monocytes) were observed. In addition, neither group exhibited noticeable fibrotic encapsulation for 5 months (Fig. 2b, c).

To comprehensively verify the biocompatibility of the constituent materials, we conducted both 3-{4,5-dimethylthiazol-2-thiazolyl}-2,5-diphenyl-2H-tetrazolium bromide (MTT) assay and single-cell gel electrophoresis (Comet) assay to confirm cytotoxicity and genotoxicity, respectively. We cultured human fibroblasts (ATCC, CRL−1502) on the surface of the component materials including the PHBV and the PHBV/PEG:ChCl, and a control culture dish for 3 d. Also, the same process was conducted for the PHBV and PHBV/PEG:ChCl membranes that underwent the HIU-triggered disintegration. As a result, the

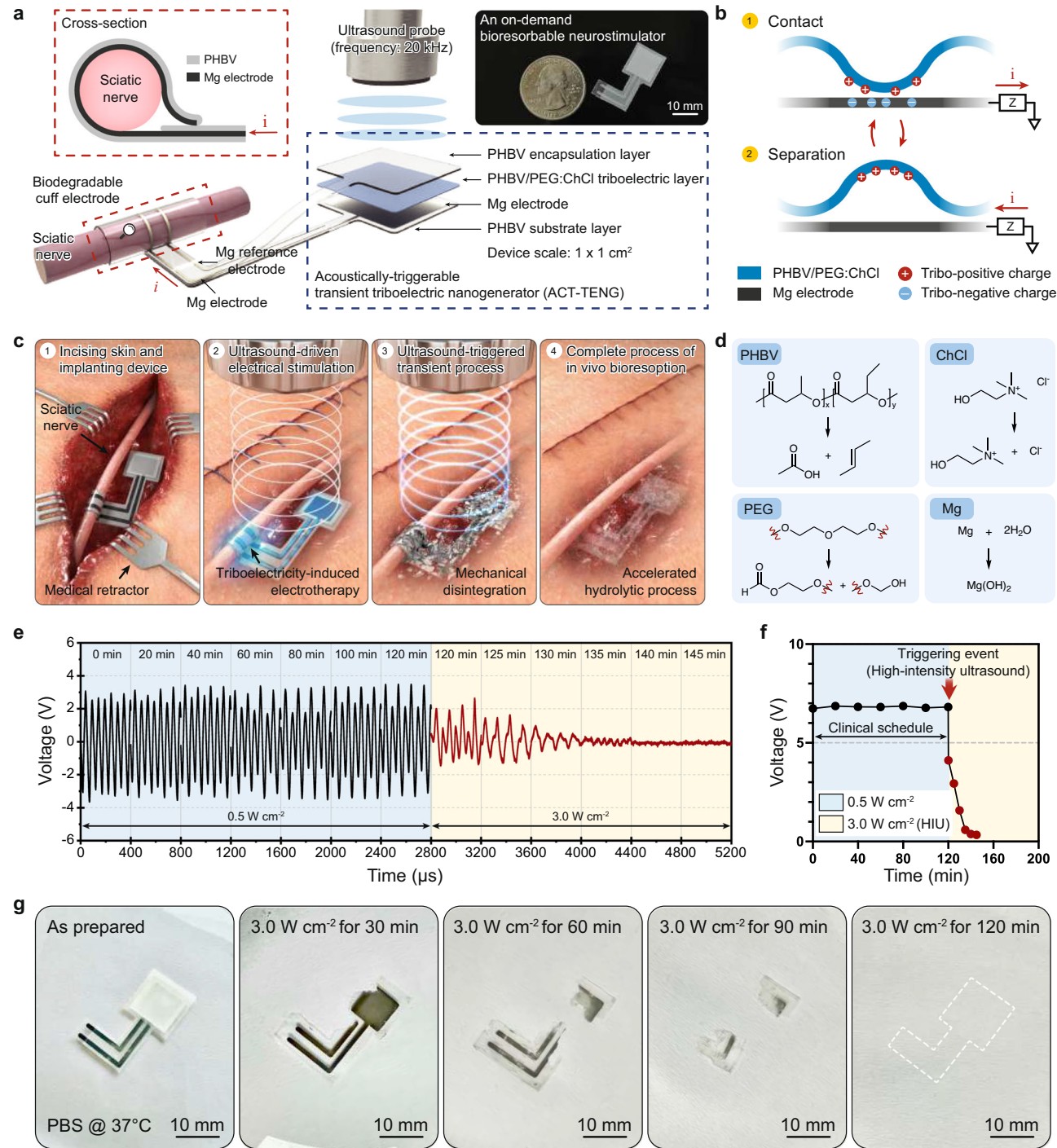

**Fig. 1 | Concept and design of the on-demand bioresorbable neurostimulator.**
**a** The overall structure of the neurostimulator with an expanded view of ACT–TENG and a bioresorbable cuff electrode. The inset displays a photograph of the device. **b** Schematic of the ACT–TENG operating mechanism. **c** Proposed clinical protocol of peripheral nerve electrotherapy using the on-demand bioresorbable neurostimulator. **d** Hydrolytic degradation processes of the constituent materials. **e** On-demand transience demonstration with long-term electrical characterization for the output of ACT–TENG. **f** The summation plots of (**e**). HIU represents high-intensity ultrasound. **g** HIU-triggered transient performance of the device immersed in diluted PBS solution (pH 7.4).

relative cell viability of human fibroblasts on each material was greater than 85% within 3 d, regardless of the presence of HIU triggering events. These results imply that there is no cytotoxicity for the constituent materials and the HIU-based active operation protocol (Fig. 2d). Figure 2e offers the results of the comet assay for the same groups to assess the genotoxicity. For all cases, the olive tail moment (OTM) length, indicating the extent of DNA strand breaks, was similar to the control group, and negligible, compared to the positive control group ($H_2O_2$) (Fig. 2f). Therefore, we concluded that there are no significant negative health consequences in terms of the materials or our HIU-driven on-demand protocol.

## Therapeutic effects of the ultrasound-driven triboelectric impulses

We performed in vivo experiments to investigate the therapeutic effects of the on-demand bioresorbable neurostimulator on both

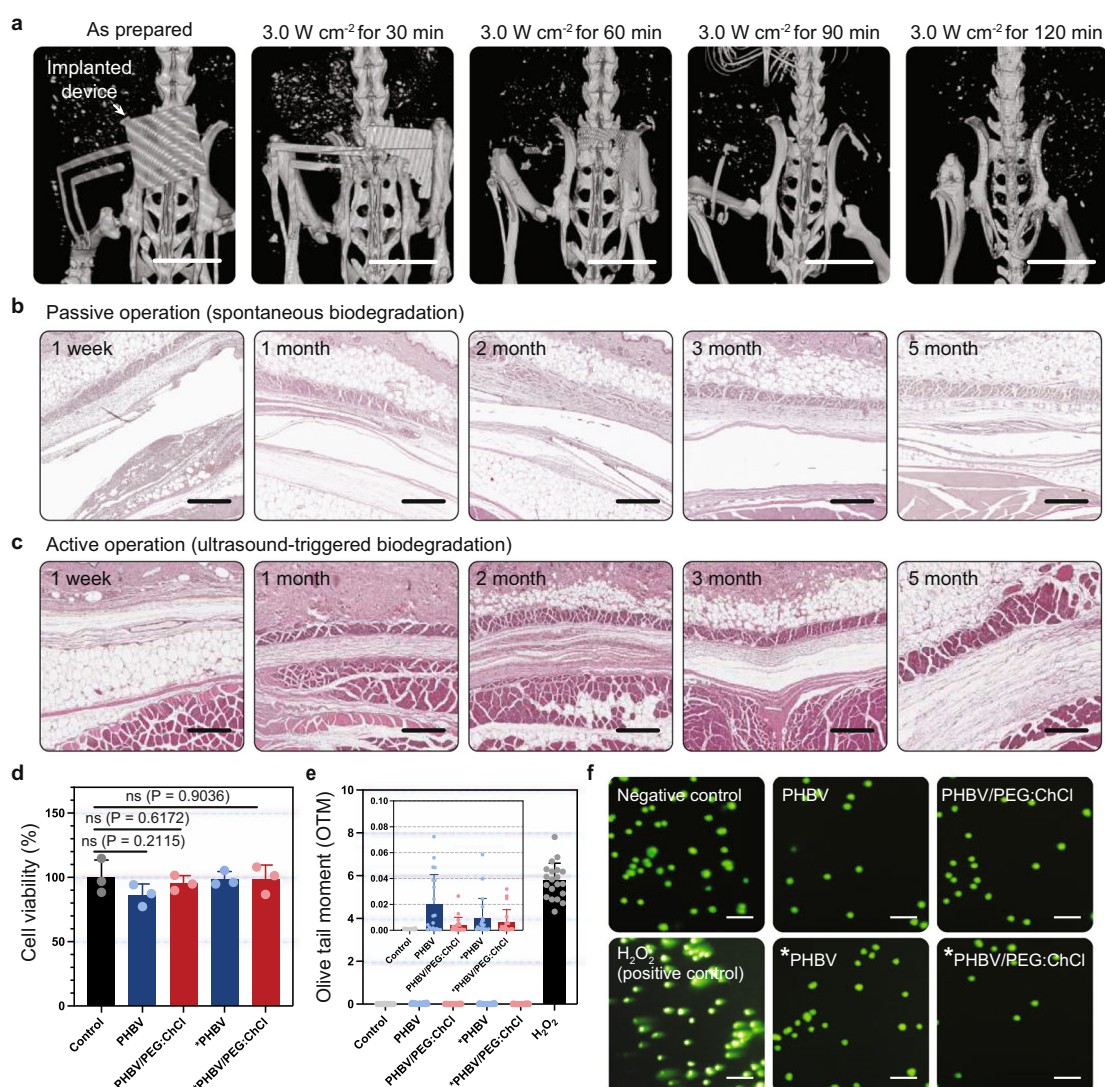

**Fig. 2 | In vivo HIU-triggered transience of the neurostimulator and its biosafety. a** Micro-CT images showing in vivo transient performances upon the HIU-driven triggering event (scale bar: 10 mm). **b, c** H&E-stained tissue images exhibiting inflammation responses with spontaneous biodegradation (**b**) and HIU-triggered transience (**c**). **d** Relative cell viability of human fibroblasts (CRL-1502) on the as-prepared and HIU-triggered materials ($n = 24$ for each group). The asterisk mark (*) indicates the HIU-induced degraded materials. Data are presented as mean values ± SD (standard deviation). The scale bar is equivalent to 100 μm. **e, f** The olive tail moment (OTM) of the as-prepared and HIU-triggered materials ($n = 20$ for each group) (**e**) and their fluorescent images (**f**), measured by comet assay. The asterisk mark (*) indicates the HIU-induced degraded materials. Data are presented as mean values ± SD. The scale bar is equivalent to 100 μm. $P$-values are evaluated through a two-sided $t$-test; ns = non significant.

compression peripheral nerve injury and CMT1A mouse models. To introduce electrical stimulation events (ESE), the ACT−TENG was implanted underneath the mouse dermis, and the connected bioresorbable cuff electrode was placed around the sciatic nerve (Fig. 3a, Supplementary Fig. 18, and Supplementary Movies 3 and 4). A pair of Mg electrodes in the cuff secured the compressed lesion of the sciatic nerve for the peripheral nerve injury model. Meanwhile, they wrapped around the CMT1A-diseased sciatic nerve for the hereditary peripheral neuropathy model. After we kept the mouse for 3 d for the recovery of the surgical wound and biological engraftments, we applied low-intensity ultrasound (0.5 W cm$^{-2}$) to deliver electrical impulses (20 kHz sinusoidal waveform) to the secured sciatic nerve for 5 d (5 min daily) (Supplementary Figs. 19 and 20). The duration and treatment schedule of ESE was determined based on the results of in vitro cell experiments using induced pluripotent stem cell (iPSC)-driven motor neurons, in which the application of ESE resulted in remarkably improved axon growth (electric field intensity = 1.3 V mm$^{-1}$; 20 kHz sinusoidal

waveform) (Fig. 3b, c, Supplementary Figs. 21−23). The in vitro cell experiments were performed over 3 days, during which electrical stimulation was administered for a duration of 5 min each day. As the lesion size of ~2 mm in the compression injury of the sciatic nerve, it can be inferred from the in vitro results that electrical output performance of the body-implanted device is at a sufficient level to perform the in vivo electrotherapy demonstration shown in Fig. 3a (Supplementary Fig. 24). On Day 8, following 5 days of electrical stimulation applied to the sciatic nerve, nerve conduction study (NCS) experiments were conducted to assess the nerve condition (Fig. 3d, e, and Supplementary Movie 5). In the NCS experiments for the compression peripheral nerve injury model, the ESE significantly increased both motor nerve conduction velocities (MNCVs) and sensory nerve conduction velocities (SNCVs) (Fig. 3d). In addition, the ESE increased the compound muscle action potentials (CMAPs) and sensory nerve action potentials (SNAPs). The improvements in the nerve conduction velocities and the action potentials, which are key evaluation factors,

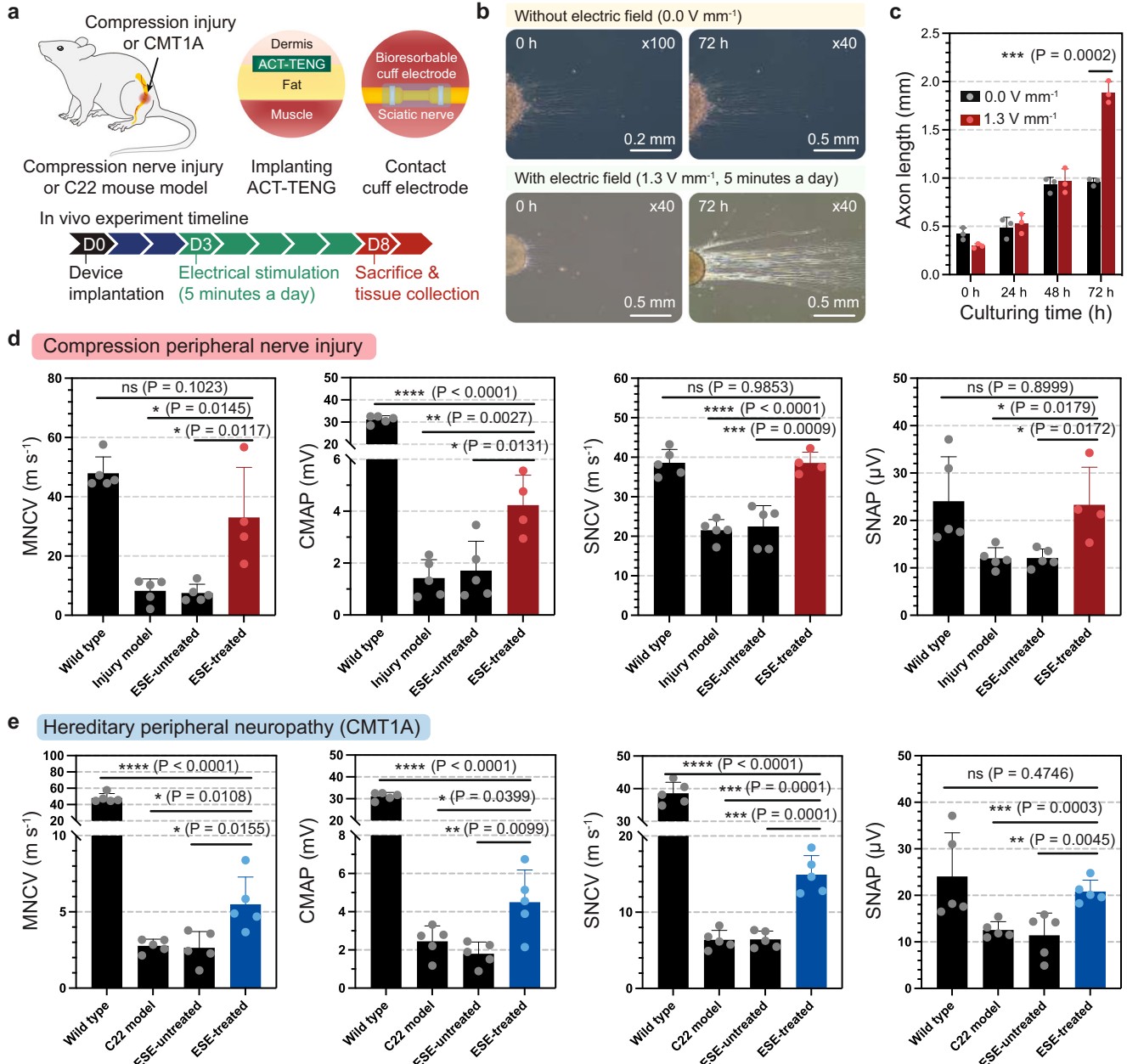

**Fig. 3 | Experimental settings for in vivo electrotherapy and the recorded NCS results for compression peripheral nerve injury model and C22 mouse (hereditary peripheral neuropathy, CMT1A) model. a** Schematic of the experimental setup for in vivo electrotherapy. **b, c** In vitro cell experiments were performed to validate the electrical stimulation conditions (e.g., amplitude, treatment duration) required for neuroregeneration. **b** Optical microscopic images of iPSC-driven motor neurons. **c** Axon length plots along the culturing time ($n = 3$ for each group). Data are presented as mean values ± SD. Based on the results of in vitro cell experiments, electrical impulses of 1.3 V mm$^{-1}$ (20 kHz sinusoidal waveform, 5 min daily) are effective for accelerating the growth of iPSC-driven motor neurons. Thus,

we adopted those electrical stimulation conditions in our in vivo demonstrations described in (**a**). **d** Recorded nerve conduction velocities and action potentials after the in vivo electrotherapy for the compression peripheral nerve injury mouse model (Wild type: $n = 5$; Injury model: $n = 5$; ESE-untreated: $n = 5$; ESE-treated: $n = 4$). Data are presented as mean values ± SD. ESE represents electrical stimulation events. **e** Recorded nerve conduction velocities and action potentials after the in vivo electrotherapy for the C22 mouse model (Wild type: $n = 5$; C22 model: $n = 5$; ESE-untreated: $n = 5$; ESE-treated: $n = 5$). Data are presented as mean values ± SD. *P*-values are evaluated through two-sided *t*-test; *$P < 0.05$; **$P < 0.01$; ***$P < 0.001$; ****$P < 0.0001$.

indicated that ESE highly promoted the physiological and pathophysiological recovery of injured sciatic nerves[28]. To assess the effect of ESE for CMT1A, we conducted experiments in the C22 mouse model, which is well known as the CMT1A model, and the ESE was applied with the same protocols (see details in "Methods", Supplementary Fig. 25). The NCS experiment results represent that the ESE significantly improved the nerve condition, as proven by increased conduction velocities (MNCVs and SNCVs) and action potentials (CMAPs and SNAPs), compared to other groups (Fig. 3e).

## Histopathological and statistical analyses for the ultrasound-driven in vivo electrotherapy

Next, we examined changes in the degree of myelination in response to the ESE by studying semi-thin sections and electron microscopic images (Fig. 4a). The neural tissue, which was collected after measuring the nerve condition through NCS experiments, was subjected to these histopathological examinations. For the compression peripheral nerve injury model, we confirmed that the ESE-treated group represented a higher degree of myelination than the peripheral nerve injury

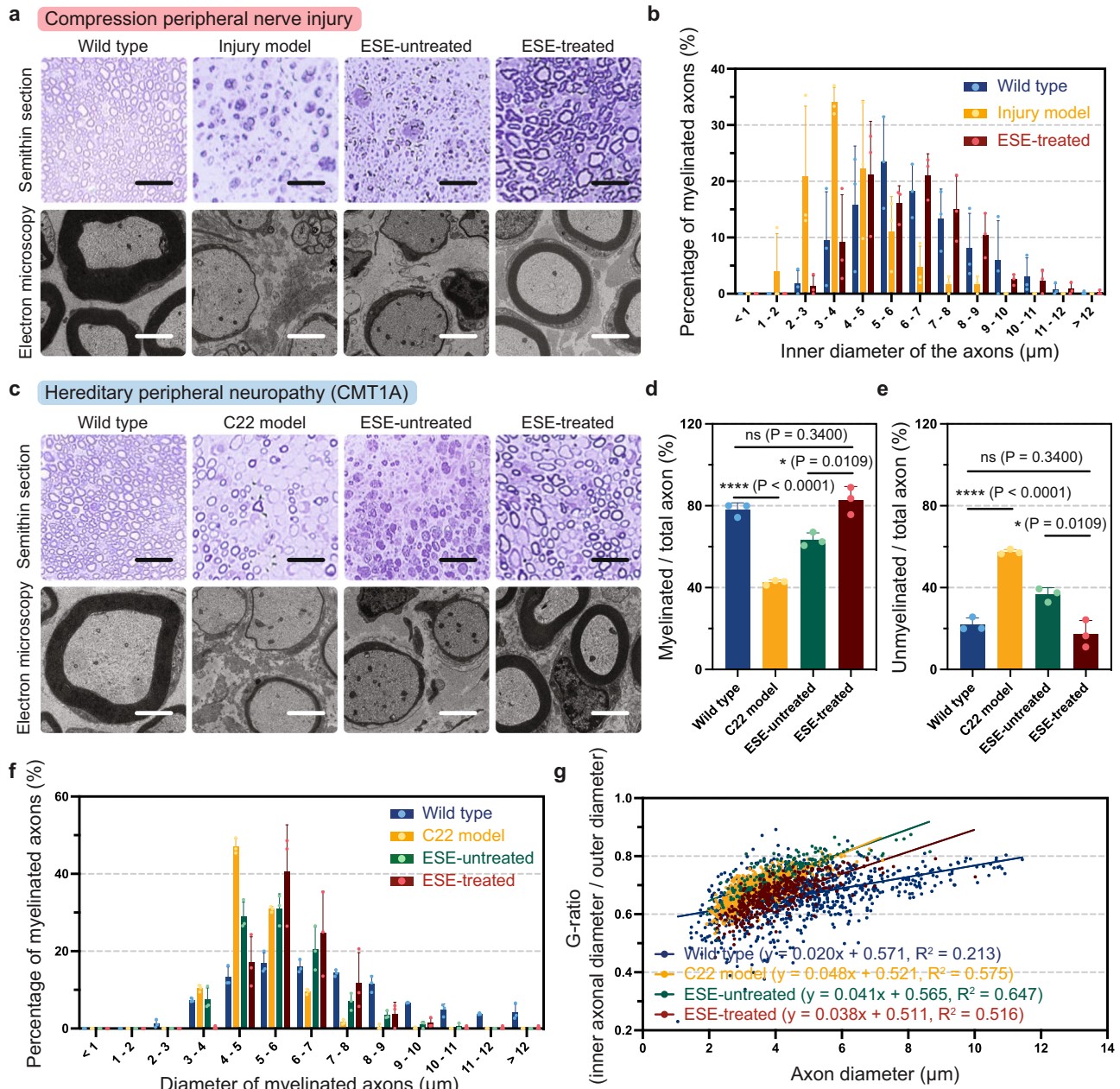

**Fig. 4 | Histopathological demonstration and statistical analyses for the therapeutic effects on both compression peripheral nerve injury and CMT1A (hereditary peripheral neuropathy). a** Semi-thin sections (scale bar: 30 μm) and electron microscopy images (scale bar: 2 μm) to observe the toluidine blue-stained sciatic nerve cross-section. **b** Histogram representing the inner diameter distribution of the myelinated axons (Wild type: $n = 3$; Injury model: $n = 3$; ESE-treated: $n = 3$). Data are presented as mean values ± SD. **c** Histopathological demonstration using semi-thin (scale bar: 30 μm) and electron microscopy images (scale bar: 2 μm) of the sciatic nerve cross-section. **d, e** Percentage of the myelinated axons (**d**) and the unmyelinated axons (**e**) (the myelinated axons refer to the axons with a diameter greater than 5 μm) (Wild type: $n = 3$; C22 model: $n = 3$; ESE-untreated: $n = 3$; ESE-treated: $n = 3$). Data are presented as mean values ± SD. **f** Histogram displaying the diameter distribution (Wild type: $n = 3$; C22 model: $n = 3$; ESE-untreated: $n = 3$; ESE-treated: $n = 3$). Data are presented as mean values ± SD. **g** Scatterplots representing the correlation of g-ratio and axon diameter (Wild type: $n = 900$; C22 model: $n = 800$; ESE-untreated: $n = 200$; ESE-treated: $n = 416$). P-values are evaluated through a two-sided t-test; ns = non significant; *$P < 0.05$; ****$P < 0.0001$.

group. In addition, we conducted a statistical analysis on the distribution of axonal diameter for each group (Fig. 4b). Note that the ESE treatment resulted in an increased number of myelinated axons as the average diameter of the axons increased, and the distribution curve shifted to the right than the compression peripheral nerve injury group.

Notably, we successfully demonstrated that the ultrasound-driven ESE promotes the recovery of the sciatic nerve in the C22 mouse model. Figure 4c expresses the myelinating effect of ESE treatment, implying its therapeutic effect. The statistical analyses displayed in Fig. 4d, e

describe that the ESE-treated group exhibited a significantly higher percentile fraction of myelinated axons, compared to the control C22 mouse model and the ESE-untreated group. In particular, the ESE-treated group represented a comparable level of myelination with the wild type (WT). The right-shifted distribution curve of the diameter of myelinated axons proved that the ESE effectively promoted myelination of the axons (Fig. 4f). To further validate the remyelination by the ESE, we composed a scatterplot of the g-ratio of axons (defined as the ratio of inner diameter to outer diameter) in correspondence to axon diameter and derived the slope of the linear trend lines (Fig. 4g). As a

result, the ESE-treated group showed lower slope compared to the control C22 group and the ESE-untreated group, indicating its higher extent of myelination. These statistical demonstrations support that the kHz frequency electrical stimulation results in the myelination of the neuronal axons for both compression peripheral nerve injury and hereditary peripheral neuropathy including CMT1A.

In this study, we undertook comprehensive investigations into the therapeutic impact of kilohertz-frequency stimulation on neuroregeneration. We postulated that the observed effects may be attributed to a sequence of events involving an elevated calcium influx, a phenomenon recognized for its pivotal role in axon growth through various potential pathways[29]. These pathways encompass potential mechanisms such as electroporation, modulation of intracellular signaling cascades, and the activation of voltage-gated calcium channels.

## Discussion

This work represents an on-demand bioresorbable neurostimulator that provides reliable electrotherapy for treating peripheral neuropathies with its manageable device lifetime. This was realized through the ultrasound-mediated transience of the neurostimulator, induced by the variation in acoustic pressure level according to the incident ultrasound intensity. At a low-intensity ultrasound (less than $1.0 \, W \, cm^{-2}$), the transient process of the device is not triggered, thereby generating consistent electrical impulses that promote myelination of the sciatic nerve. Meanwhile, the HIU (more than $3.0 \, W \, cm^{-2}$) triggers the in vivo disintegration of the neurostimulator, providing on-demand device elimination according to the clinical needs and situation. Moreover, we validated the biocompatibility of the on-demand bioresorbable neurostimulator, with no noticeable immune responses, cytotoxicity, or genotoxicity (Supplementary Movie 6). We also observed that the HIU-induced disintegrated particles did not provoke any adverse health consequences. The therapeutic effect of the neurostimulator was investigated by nerve conduction study and histopathological analyses. The high-frequency electrical stimulation (20 kHz, 7.76 V) introduced distinct improvements in the conduction velocities and action potentials for both the compression peripheral nerve injury and hereditary CMT1A mouse models. The statistical analysis, based on semi-thin section and electron microscope imagery, also proved that the electrical stimulation results in the myelination of axons for both disease models.

Bioresorbable electronics are of significant interest in the fields of biomedical engineering and medicine, as they eliminate the need for device extraction surgeries after fulfilling their intended clinical purposes. However, conventional devices face challenges regarding their transient mechanism, as the transient rate is solely determined by the material properties and dimensions. This can result in either premature degradation, failing to ensure the desired treatment duration, or prolonged device lifetime, leading to potential negative health consequences due to the presence of device residues. Thus, there is a growing need for protocols that can accelerate the transient rate, allowing for controlled and timely device dissolution. In this study, we have developed an on-demand neurostimulator that can actively control the lifespan of the implanted bioresorbable device using biosafe ultrasound at the desired time point. Through this work, we anticipate the personalized treatment of patients without the need for device removal surgeries, as the operation of the bioresorbable neurostimulator can be tailored according to the patient's treatment circumstances.

## Methods

### Materials synthesis and preparation

Poly(3-hydroxybutyrate-co-3-hydroxyvalerate) (PHBV, PHV content 2%) was obtained from Goodfellow, and polyethylene glycol (PEG, average Mn ~20,000 g mol⁻¹), (2-Hydroxyethyl)trimethylammonium

chloride (ChCl), and anhydrous chloroform were obtained from Sigma-Aldrich. The PHBV film preparation began with dissolving 1.3 g PHBV in 20 mL chloroform (Sigma-Aldrich) in a dried glass vial. The solution was stirred at 70 °C using a hot plate for about 2 h, until full dissolution. The solution was drop-cast on a glass mold and dried in ambient conditions, and the dried film was gently peeled off. To prepare the PHBV/PEG:ChCl composite film, 1.3 g PHBV and 186 mg PEG were dissolved in 20 mL chloroform in 7:1 weight ratio, followed by mixing ChCl (21.4, 42.7, 213.5, 427.1, and 640.6) mg for (0.005, 0.010, 0.05, 0.1, and 0.15) molality. The solution was stirred and drop-cast in the same way. The thickness of both films was defined by the volume and area of the solutions. The above films were kept in diluted PBS solution (Biowest) before use, to remove potentially cytotoxic or genotoxic residual reagents.

### Device fabrication

Supplementary Figure 1 provides a schematic of the device fabrication process. Our neurostimulator comprises an ACT−TENG and a cuff electrode that are integrated into a single structure. A PHBV substrate with a thickness of (30−50) μm was cut into the desired shape, and magnesium (Mg) was deposited on the substrate by electron beam evaporation with masking to fabricate the electrodes and leads of our device. The electrode of ACT−TENG has a square shape of 8 × 8 mm dimensions. A square (30−50) μm thick PHBV/PEG:ChCl composite film of 9 × 9 mm dimensions was placed on the electrode of the ACT−TENG to fully cover it, and another PHBV film was integrated to cover the whole area of the device through hot-pressing aiming at encapsulation.

### Materials analysis

The surface morphology of PHBV and PHBV/PEG:ChCl was observed by field emission-scanning microscopy (FE−SEM, Jeol Ltd., JSM-6701F) and optical microscopy (SAMWON, XT−VISION−UM2). The surface potential was investigated through a KPFM system with a Pt/Cr-coated silicon tip and lock-in amplifier (Stanford Research, SR830). The surface potential of the sample was estimated using highly ordered pyrolytic graphite (HOPG) as a reference sample. The crystal properties and thermal stability were measured by XRD (Bruker−AXS, D8 Discover) and DSC (SEICO INST., DSC7020), respectively. Mechanical tests were performed by a universal testing machine (Instron, Instron 5844) to obtain the strain−stress curve, and Young's modulus of the component materials. We used a specific gravity scale (AND, GR-200D) to measure the density of the membranes.

### Analysis of in vitro transient processes

To assess the in vitro transient processes with passive operation, each material and device was immersed in diluted PBS solution (Biowest, pH 7.4) at 37 °C. At the desired time points, the mass was measured using an electronic scale (RADWAG, AS 220.R2 Plus), and a series of photographs were taken to demonstrate the in vitro transient processes over time. In the case of active operation, high-intensity ultrasound (HIU, $3.0 \, W \, cm^{-2}$) was applied to each material and device immersed in the diluted PBS solution, and changes in mass and morphology were obtained by the same process. The ultrasound intensity was set based on the standards outlined by the International Electrotechnical Commission (IEC) for ultrasound-driven therapy equipment (IEC standard 60601-2-5). This standard establishes a maximum threshold for the effective intensity, defined as the ratio of acoustic output power to effective radiating area, which is set at $3.0 \, W \, cm^{-2}$.

We measured pH changes by accelerated hydrolysis of our on-demand bioresorbable neurostimulator to assess the biological safety of on-demand transience by HIU-driven triggering events. The device was immersed in diluted PBS solution in a dried glass vial, and HIU was applied to the film with a distance of 5 mm for (30, 60, 90, and 120) min. At each time point, pH was measured using a pH meter (SANXIN, SX620).

## Electrical characterization

The output voltage was measured by an oscilloscope (Tektronix, DPO3052) and a probe (Tektronix, P5100A) with a 40-megohm input impedance. The output current was measured using a low noise current amplifier (FEMTO, DLPCA-200) with a 1-megohm input impedance, which was connected to the oscilloscope. For in vitro electrical characterization, low-intensity ultrasound or HIU was applied to the ACT-TENG immersed in deionized water at a 5 mm distance from the ultrasound probe.

## MTT assay, XTT assay, and comet assay

To perform the MTT assay, human fibroblasts (ATCC, CRL-1502; Designation: WS1; Product category: human cells; Organism: Homo Sapience, human; Cell type: fibroblast; Morphology: fibroblast) and culture medium were added after placing 5 × 5 mm experimental films into each well of 96-well plates (10,000 cells/well, final volume: 100 μL). Then, we incubated the fibroblasts for each period of (24, 48, and 72) h. Subsequently, we removed the culture medium from the 96-well plates (medium suction), and added 50 μL MTT solution (Thermo Fisher Scientific, 5 mg mL$^{-1}$) and 50 μL culture medium to each well. After incubation for 2 h at 37 °C, we removed the MTT solution and the culture medium. We cleaned the samples by rinsing them with diluted PBS solution and added 150 μL solubilization solution (dimethyl sulfoxide, Sigma-Aldrich) to dissolve formazan crystals. Finally, we transferred 100 μL of the solution to another 96-well plate and measured the absorbance to evaluate the viability (OD = 590 nm).

We purchased Cell Proliferation Kit II (XTT) (Sigma-Aldrich, 11465015001) to conduct the XTT assay. Human fibroblasts (ATCC, CRL-1502) and culture medium were added upon the presence of experimental films (5 × 5 mm) into each well of 96-well plates (10,000 cells/well, final volume: 100 μL). They were incubated for 3 days at 37 °C with 5% $CO_2$ concentration. Before measuring the absorbance, we added 50 μL of XTT labeling mixture per well and incubated for 18 h at 37 °C with 5% $CO_2$. Then, the absorbance of the samples was measured using a microplate (ELISA) reader to evaluate the cell viability (OD = 490 nm).

To perform the comet assay, we first prepared a lysis buffer (Thermo Fisher Scientific), alkaline solution (SAMCHUN Chemical), and tris-borate ethylene-diamine-tetraacetic acid (TBE) electrophoresis solution (Thermo Fisher Scientific). We placed test films onto the slide in the wells. We melted the agarose gel until liquefied, and added 75 μL agarose gel to the slide to form the base layer. Then, we kept the slide at 4 °C for 15 min. We attached human fibroblasts to agarose gel (1/10 ratio) and transferred 75 μL of the mixture on top of the base layer. Then, we kept the slide at 4 °C for 15 min. We immersed the slide in pre-chilled Lysis buffer (25 mL) for 60 min at 4 °C in dark conditions and the pre-chilled Alkaline solution (25 mL) for 30 min at 4 °C in dark conditions. Subsequently, we immersed the slide in pre-chilled TBE electrophoresis solution for 10 min. To perform electrophoresis, we filled the well with cold TBE electrophoresis solution until the solution covered the slide, and applied an electric field of 0.1 V mm$^{-1}$ to it for 15 min. After electrophoresis, we rinsed the slide three times with pre-chilled DI water and 70% ethanol solution. After the slide was dried, we added 100 μL diluted Vista Green DNA dye, and incubated it for 15 min at room temperature. Finally, we observed the slide with epifluorescence microscopy using a FITC filter and measured the olive tail moment (OTM).

## Preparation of iPSC-driven motor neuron

This study was approved by the Institutional Review Boards of Samsung Medical Center at Sungkyunkwan University (2021-04-053-001). Several key steps were involved to obtain human iPSCs (induced pluripotent stem cells). First, skin tissue is obtained through a skin biopsy from a single participant, a 53-year-old female,

accompanied by the collection of pertinent information from the clinical records (e.g., genetic, clinical, and electrophysiological test results). The skin biopsy was performed under suitable conditions, with circular tissue specimens of 3 mm diameter extracted from the ankle regions of the participant. We acquired consent to publish information that identifies this participant.

These skin tissues were then used to procure fibroblasts. To procure fibroblasts, skin tissue samples were cut into small pieces and placed in culture dishes. We observed that the cells were growing in the culture dishes after 3 days. During the second week of culture, fibroblasts had proliferated to approximately 80% of the culture area. Subculturing was performed more than twice to isolate and culture pure fibroblasts.

Subsequently, we used the CytoTune-iPS 2.0 Sendai Reprogramming Kit (Invitrogen Co. Ltd.) and a set of transcription factors to reprogram fibroblasts. The kit was utilized following the instructions from the manufacturer to induce the expression of four reprogramming factors: OCT3/4, KLF4, SOX2, and cMYC. After 15 days, cell colonies with an embryonic stem cell-like appearance were isolated. The iPSCs were then expanded under feeder-free conditions using mTeSR medium (STEMCELL Technologies).

The pluripotency and differentiation potential of the iPSCs were confirmed through analytic methods encompassing fluorescence staining techniques (OCT4, SSEA4, Tra-1-60 confirmation) and pluripotent cell isolation procedures (SSEA4 confirmation). Before iPSC-driven motor neuron preparation, wild-type human iPSCs in Essential 8 medium (Thermo Fisher Scientific, Waltham, MA, USA) in a sterilized vial were stored under cryogenic conditions in liquid nitrogen. To culture iPSCs, the cell vials were incubated in a water bath at 37 °C for 3 min. Then, 1 mL cell suspension was transferred to a 15 mL centrifuge tube and topped up to 5 mL with Essential 8 medium containing 10 μM Y-27632 (Tocris Bioscience), a selective Rho-kinase inhibitor. The cell suspension was centrifuged for 3 min at 300×g. After aspiring the supernatant, the cells were resuspended in fresh Essential 8 medium and Y-27632. The human iPSCs were routinely cultured on 6-well culture plates coated with Matrigel. Matrigel (Thermo Fisher Scientific, Waltham, MA, USA) diluted 1 to 60 in DMEM/F12 medium (Thermo Fisher Scientific) was poured onto dishes of 10 cm diameter and incubated at 37 °C/5% $CO_2$ overnight. Prepared Matrigel-coated plates were then aspirated and rinsed using Dulbecco's phosphate-buffered saline (DPBS, Thermo Fisher Scientific). The cell suspension was plated on the prepared Matrigel-coated plates and incubated for culturing. Y-27632 was removed from the culture medium on the first day of plating and the cells were fed daily with fresh Essential 8.

For differentiation of human iPSCs to motor neurons, cultured human iPSCs were transferred onto Matrigel-coated plates as described above and incubated at 37 °C/5% $CO_2$ in Essential 8 until the confluency reached about 80%. At this point, human iPSCs were differentiated into regionally unspecified neural progenitor cells using a monolayer differentiation method[30]. These cells were then passaged onto Matrigel-coated plates and incubated for motor neuron differentiation[31].

## Analysis of in vivo transient processes

All studies on mice were approved by the Institutional Animal Care and Use Committee (IACUC) of Samsung Medical Center (20210318001). The 10-week-old C57BL/6J and CrlOri:CD1(ICR) females were prepared and provided with a standard diet and water ad libitum, and housed in a temperature-controlled (22 ± 2) °C and humidity-controlled (44–56)% environment with a 12 h light–dark cycle. The analysis of in vivo transient processes began with anesthesia to a mouse model (10 weeks, C57BL/6) by inhalation of (2–5)% isoflurane (Hana Pharm Co., Ltd.), and maintained with 2% isoflurane. The mouse model was then fixed in a prone position, and the dorsum was shaved and cleaned using a povidone-iodine prep pad (Green Pharm Co., Ltd.). The ACT-TENG was

implanted under the dermis of the mouse model after UV sterilization, and the cuff electrode was wrapped around the sciatic nerve by suturing. The invasion was closed with skin and muscle sutures, and the mouse model was left for 3 days. For the HIU-driven triggering event, anesthesia was conducted for the mouse model in the same way. We applied HIU to the mouse model at a distance of 5 mm from the ultrasound probe for the desired duration, and then obtained micro-CT images (Bruker, Skyscan 1276) under the following conditions: source voltage of 70 kV, source current of 200 µA, the applied filter of Al (0.5 mm), and scanning rotation step of 0.400 degrees. NRecon software (Bruker) was used for the reconstruction process of the micro-CT images. Then, CTAn software (Bruker) was used to analyze the reconstructed images. Finally, CTVox software (Bruker) was used to implement a volume-rendering process for the analyzed images.

### In vivo electrical characterization
For in vivo electrical characterization, anesthesia was conducted on a mouse model (10-week, C57BL/6) by the inhalation of isoflurane (Hana Pharm Co., Ltd.). Then, the mouse model was fixed in a prone position and the dorsum was shaved, followed by topical application of a povidone-iodine prep pad (Green Pharm Co., Ltd.). The ACT–TENG was implanted under the dermis of the mouse model after UV sterilization, and the invasions were closed with skin sutures, while the minimized lead wire was left outside to reduce pain and inflammation. The surgical wound was dressed in sterilized gauze, and the mouse model was left for 3 days, aiming at the recovery of the surgical wound and biological engraftments of the ACT–TENG. Then, anesthesia was conducted on the mouse model again. We applied ultrasound to the mouse model at a distance of 5 mm from the ultrasound probe and measured the output voltage and current through direct coupling the lead wire to oscilloscope probes, and the mouse model was kept under an infrared heater to maintain the body temperature.

### In vivo biocompatibility analysis
Anesthesia and implanting of the neurostimulator were performed with the abovementioned processes. We harvested subdermal tissue of the implanted area at the desired time points and carried out hematoxylin and eosin staining (H&E staining) to observe inflammation.

### Animal preparation (the compression peripheral nerve injury mouse model)
The 10-week-old C57BL/6J females were prepared and provided with a standard diet and water ad libitum, and housed in a temperature-controlled ($22 \pm 2$) °C and humidity-controlled (44–56)% environment with a 12 h light–dark cycle. Anesthesia was conducted on the mouse model by the inhalation of (2–5)% isoflurane, and maintained with 2% isoflurane. Then, the mouse model was fixed in a prone position, and the skin on the lateral surface of the left thigh was shaved, followed by topical application of a povidone-iodine prep pad. A single, small skin incision was made at mid-thigh level with fine scissors, followed by dissection using the dissection scissors between the biceps femoris muscle and the gluteus muscle, to expose the sciatic nerve. To induce a compression injury with a controlled level of pressure, a specific type of surgical forceps, called the HALSEY needle holder (smooth jaws, total length: 12.5 cm) was employed. The needle holder comprised a total of three stages of holder drivers, of which the second stage was utilized to apply compression on the sciatic nerve for 5 s. The in vivo electrical stimulation was conducted after implanting the neurostimulator, closing the invasion with skin and muscle sutures, and surgical wound recovery for 3 d.

### Animal preparation (C22 mouse model)
The C22 mice [B6; CBACa-Tg (*PMP22*) C22Clh/H], which harbor seven copies of a human *PMP22* transgene (Huxley et al., 1996), were

obtained from MRC Harwell (Oxfordshire, UK; IMSR Cat# HAR:784, RRID: IMSR_HAR:784). The C22 and wild-type (WT) mice were produced by in vitro fertilization using 10-week-old C57BL/6J females (OrientBio, Republic of Korea) and 12-week-old C22 males (Macrogen, RRID:SCR_014454). Animals were provided with a standard diet and water ad libitum and housed in a temperature-controlled ($22 \pm 2$) °C and humidity-controlled (44–56)% environment with a 12 h light–dark cycle. Male C22 or WT littermates were used for experiments when they were 3 weeks old.

### In vivo electrical stimulation
The mice were randomly divided into the desired groups. Anesthesia and implanting of the neurostimulator were performed with the abovementioned processes to prepare the ESE-treated group. For in vivo electrical stimulation, anesthesia was conducted on the mouse model. We applied 0.5 W cm$^{-2}$ ultrasound to the mouse model at a distance of 5 mm from the ultrasound probe for 5 d at 5 min daily. After every in vivo electrical stimulation, the mouse model was kept under an infrared heater to maintain the body temperature. No devices were implanted in the wild type.

### Electrophysiological status assessment
The nerve conduction study (NCS) was performed using a Nicolet Viking Quest device (Natus Medical, San Carlos). Mice were anesthetized by inhaling isoflurane for the duration of the procedure. To measure motor nerve conduction velocity (MNCV) and compound muscle action potential (CMAP), the active recording needle electrode (cathode) was placed onto the gastrocnemius muscle with the reference electrode (anode) on its tendon. The stimulating cathode was perpendicularly inserted approximately 2 mm under the skin without direct contact with the nerve, at the position of 6 mm proximal to the recording electrode in the midline of the posterior thigh, and 6 mm proximally in the medial gluteal region, to obtain distal and proximal responses, respectively. The stimulating anode was subcutaneously placed in the midline over the sacrum. A surface electrode as a ground electrode was placed on the mouse tail. Finally, single square-wave pulses of 0.1 ms duration were delivered to obtain the conduction signal. The amplitudes of MNCV and CMAP at supramaximal stimulation were determined by an independent examiner who was blinded. To measure the sensory nerve conduction velocity (SNCV) and sensory nerve action potential (SNAP), stimulating cathode and anode were perpendicularly inserted into the distal tail at a distance of 5 mm, and active recording needle electrodes were placed onto the proximal tail at a distance of 5 mm. The distance between the stimulating electrodes and the active recording needle electrodes was 30 mm. A surface electrode as a ground electrode was placed in a middle position between the stimulating electrodes and the active recording needle electrodes. Finally, single square-wave pulses of 0.1 ms duration were delivered to obtain the conduction signal. The amplitudes of SNCV and SNAP at supramaximal stimulation were determined by an independent examiner who was blinded.

### Histopathological examinations
Sciatic nerves were biopsied from the mouse models. The specimens were fixed overnight with 2.5% glutaraldehyde (Sigma-Aldrich) in 4% paraformaldehyde solution (Biowest) at 4 °C. After incubation in 1% OsO$_4$ (Sigma-Aldrich) for 1 h, the specimens were dehydrated in an ethanol series, passed through propylene oxide (DAEJUNG), and embedded in epoxy resin (Epok 812). Tissues were cut into semi-thin (1 µm) sections, and stained with toluidine blue (Sigma-Aldrich) for (5 to 10) s. Semi-thin sections were imaged using a BX51 upright microscope (Olympus) and analyzed using Cell Sense (Olympus). Ultrathin sections (65 nm) were collected on 200 mesh nickel grids, and stained with 2% uranyl acetate (Sigma-Aldrich) for 15 min, and lead citrate (Sigma-Aldrich) for 5 min. These specimens were observed by Hitachi

HT7700 electron microscopy at 100 kV. The axon diameter was determined using the Zeiss Zen2 software (Carl Zeiss). The inner/outer diameters of axons were measured using an image processing program (ZEISS, ZEN Microscopy Software). To obtain the percentile fractions of myelinated axons, the number of myelinated/unmyelinated axons was manually counted (an axon of an outer diameter larger than 5 μm was considered as myelinated).

## Statistics and reproducibility

Prism 9 software (version 9.5.0, GraphPad) was used to evaluate the statistical significance of all comparative data. Statistical significance between groups was determined by unpaired Student's $t$-test (two-tailed), with the thresholds of $*P < 0.05$, $**P \leq 0.01$, $***P \leq 0.001$, and $****P \leq 0.0001$. In order to validate the inflammatory response throughout the biodegradation process, five mice for each operation mechanism were sacrificed, one at a time during each designated time period, to obtain H&E-stained tissue images (Fig. 2b, c).

## Reporting summary

Further information on research design is available in the Nature Portfolio Reporting Summary linked to this article.

## Data availability

All data supporting the findings of this study are available within the article and its supplementary files. Additionally, the data used in this study are available in the Figshare Repository under accession code [https://doi.org/10.6084/m9.figshare.24270595.v1][32]. Any additional requests for information can be directed to, and will be fulfilled by, the corresponding authors. Source data are provided with this paper.

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

## Acknowledgements

S.-W.K. acknowledges support from the Nano Material Technology Development Program (2020M3H4A1A03084600), the Bio & Medical Technology Development Program (2022M3E5E9082206), the Basic Science Research Program (2022R1A3B107829, Research Leader

Program) through the National Research Foundation of Korea (NRF) funded by the Korean government (MSIT) and the YONSEI World-Class Fellow Program funded by Y.J. Lee. B.-O.C. acknowledges support from the KHIDI by the Ministry of Health & Welfare grant HR22C1363 (Korea Health Technology R&D Project), and Future Medicine 2030 Project of the Samsung Medical Center (SMX122005).

## Author contributions

D.-M.L., M.K., I.H., B.-O.C., and S.-W.K. conceptualized the ideas and identified the methodology of the experiments. D.-M.L., I.H., and B.J.P. performed the experiments including the in vivo demonstrations. D.-M.L. and I.H. analyzed and visualized the experimental data. D.-M.L. performed FEM computational simulation. D.-M.L. and M.K. wrote the manuscript. D.-M.L., M.K., H.J.K., H.-J.Y., H.R., S.H.N., H.-m.P., B.-O.C., and S.-W.K. discussed the experimental results and commented on the manuscript. B.-O.C. and S.-W.K. supervised this project.

## Competing interests

D.-M.L., M.K., and S.-W.K. are inventors on the patent (KR/ 10-2348997) and patent application (US/ 17/515,675) filed through the Sungkyunkwan University Research and Business Foundation which cover the on-demand bioresorbable neurostimulator for peripheral nerve electrotherapy used in this work. Energymining Co., Ltd. acquired the patent and patent application through technology transfer. The remaining authors declare no competing interests.
