## [Peer Review File · Nature Communications]

REVIEWER COMMENTS

Reviewer #1 (Remarks to the Author):

In this work, the authors present an implantable ultrasonic transducer based on the principle of triboelectricity. The paper's results are very interesting. The multifunctional usage of ultrasound is the major innovation point of this paper. In this study, ultrasound was used not only as a medium for energy delivery, but also as a trigger for in vivo degradation. In addition, the authors performed a significant amount of biomedical experiments, which demonstrated the application prospects of implantable triboelectric energy conversion technology.

However, there are still several comments that need to be addressed by the authors. These contents mainly include the structural design of the triboelectric device, the balance between the functionality and safety of ultrasonic power, and the biocompatibility of the device etc. The specific comments are as follows.

1. The performance comparison between the current device and the commercially-available nerve stimulators is missing. Could the authors use Bland-Altman analysis or other methods to perform such kind of comparison ?
2. The reviewer is a little bit confused on the sentence "The locally intensified acoustic pressure in the porous structure of the PHBV membrane triggers its mechanical disintegration" in Fig. S13. Could the authors provide more explanation?
3. If the unit of Fig. S13 is dB, the maximum and minimum values of acoustic pressure are 231.450 dB and 231.440 dB respectively, and the corresponding absolute pressures are 7.4736 MPa and 7.4650 MPa respectively. The difference between them is less than 0.12%. The reason why such a small acoustic pressure difference can lead to mechanical disintegration needs more detailed discussion.
4. Fig. 3b-c are obtained from the in-vitro cell experiments, is it the same for Fig. 3d-e? Considering that Fig. 3a is an in-vivo characterization model, the in vitro or in vivo methods used in subsequent experiments need to be explained in the manuscript.

5. The authors should briefly discuss the effect of the output voltage amplitude of the neurostimulator on electrotherapy for treating peripheral neuropathies. Different ultrasound power will affect the output voltage of ACT-TENG, how can the authors determine that the curative effect of 0.5W/cm² the most suitable value?

6. Supplementary Fig. 8 shows the working principle of the TENG used. The TENG appears to be a TENG in single-electrode mode. In fact, previous studies have shown that the single-electrode mode TENG is not the highest in terms of energy conversion efficiency. The charge transfer efficiency is significantly limited for this mode. The authors should add some descriptions and rationale of the TENG structure design in the text.

7. If this is the most suitable mode TENG, the authors should discuss the choice of structural parameters, such as the thickness of the Mg electrode, the distance between the electrode and the triboelectric layer. In particular, theoretically speaking, different distances between the electrode and the triboelectric layer will result in different TENG outputs. Then, if there is a certain most suitable stimulation voltage, can it be adjusted by changing the structural parameters while keeping the ultrasound power constant?

8. One of the conditions for this device to work is that the hermeticity of the device remains good. The author should provide comments on the hermeticity of this device.

9. There is no complex power management circuit in this paper. In this case, we believe that the intensity of electrical stimulation is affected by the intensity of ultrasound. The power density of 0.5W/cm² is used in the normal electrical stimulation, but the power density of degradation is 6 times of that. Is it possible that the intensity of electrical stimulation required for treatment needs to be large enough (considering the severity of the disease), but the corresponding ultrasound intensity at this time will already cause the device to start to disintegrate? Please explain this part.

10. If the above situation does not exist, how to determine the optimal ultrasonic power intensity? What this paper proposes is an innovative device with a platform nature. Is it necessary to add a circuit that adjusts the intensity of electrical stimulation to adapt to different treatment intensities and treatment frequencies? (Avoid the degradation area that exists by directly adjusting the ultrasound intensity)

11. The reviewer still has concerns about the biocompatibility of the device. On the one hand, whether the concentration of Mg will cause toxicity to the human body; on the other hand, whether the intensity of ultrasound will affect the health of the human body. The authors should provide comments on the biocompatibility of such devices.

12. HIU (more than 3.0 W/cm²) triggers the in vivo disintegration of the neurostimulator, providing on-demand device elimination according to the clinical needs and situation. However, is intensity over 3W/cm² safe for animals or people?

Some Minor Comments:

1. A more detailed introduction part will help readers grasp the technical basis of this article. The author may consider supplementing relevant literature on ultrasound based TENG and nerve electrical stimulation therapy, and compare the advantages and disadvantages of RF-based electrotherapy and ultrasound-based electrotherapy.

2. In the in vitro experiment part, PBS solution was used to verify the insolubility of the device itself and the degradation performance under ultrasonic triggering. But the Supplementary Fig. 10 shows that the experimental setup is under deionized water environment, not PBS. Can this deionized water environment be replaced with a PBS solution?

3. The unit of Fig. S13 is not marked.

Reviewer #2 (Remarks to the Author):

The article proposes an on-demand bioresorbable neurostimulator based on triboelectric nanogenerator. An ultrasound source enables electrical stimulation and rapid device elimination. However, the evaluation of important performance is absent, the underlying mechanism of high-frequency electrical stimulation on nerve regeneration is not clear, and the histology studies demonstrating the therapeutic effects are not sufficiently convincing. These critical issues greatly diminish the quality of the work and cannot warrant publication on Nature Communications.

1. The Mg electrode adopted in the work is quite thin, with a thickness of only 100 nm. This renders it susceptible to rapid degradation when exposed to physiological solutions, which can be accelerated further by even low-intensity ultrasound. However, the authors did not provide the evaluation of the operational lifetime of the device in aqueous environments. The authors should provide evidence that the device can perform over the long term to offer adequate therapeutic support. In Fig. 1, the

operational window is shown up to only 120 minutes; however, this time frame does not suffice for effective nerve regeneration.

2. The discussion regarding the underlying mechanism of the work appears to be weak. Typically, low-frequency electrical stimulation (below 500 Hz) is employed for neural activation, whereas high-frequency stimulation is utilized for conduction block to alleviate pain (Science Advances 8, eabp9169, 2022). Given that high-frequency stimulation (~20 kHz) sustains depolarization and helps to suppress action potentials, it is unclear how this technique can promote nerve regeneration which needs activation.

3. What is the mechanism behind the acceleration of PHBV degradation with high-intensity ultrasound? Does the polymer undergo hydrolysis or disintegration? Moreover, what are the benefits of triggered degradation? The accelerated degradation could result in the intense release of acidic monomers and metallic ions, causing toxic effects.

4. The assessment of the regeneration of sciatic nerves is not rigorously performed and presented. Evaluation through histological studies is highly sensitive to the location of the measurement and careful identification of the right location is critical. The histological results of remyelination in Fig. 4 are not convincing. The authors show that the remyelination of injured nerve with ESE treatment is comparable to that of the wild type after the regrowth of only 8 days, which is impossible. Refer to previously published papers, (for example, Nature Medicine volume 24, pages1830–1836, 2018), it takes around 8 weeks to achieve high degree of remyelination and the recovery of EMG amplitude, on dissected sciatic nerves, which is the same injured model adopted in the current work. In fact, the CAMP of the ESE treated group is much lower compared to that of the wild type group (Fig. 3d) is an indication of low degree of remyelination.

5. Other issues:

- In Fig. S4b, the labeling of hydrogen bond is incorrect. There should not be hydrogen bonding between CH₃ and O, as hydrogen is specifically referring to the intermolecular force when a hydrogen atom bonded to a strongly electronegative atom (such as F, N, O) exists in the vicinity of another electronegative atom with a lone pair of electrons.

- The statistical analysis is confusing in Fig. 3d, e. There is obviously significant difference between the wild type and ESE-treated groups, but the significance is not labeled.

Reviewer #3 (Remarks to the Author):

This is an interesting paper describing a dissolvable stimulator used to deliver electrical stimulation to peripheral nerves. The authors have shown promising results in this paper. While my area of expertise lies in peripheral nerve stimulation and injury, I do have some comments about other aspects of the paper before I detail my comments on the stimulation and nerve injury itself.

General comments about stimulator dissolution

- With respect to accelerating the dissolution of the stimulator, why do the authors believe that no long term residues exist once their stimulator has undergone the dissolving triggering events?
- Does HIU damage any tissue or mechanical deform any tissue?

Biocompatibility comments

- In terms of biocompatibility of the device, the authors chose to use the MTT method instead of the more widely used XTT method, why is that?

Additionally, typical biocompatibility is done to an ISO standard (10993-5) which documents MTT methods for cell culture and using L929 mouse fibroblast cells not human fibroblast cells.

- How did the authors come up with their biocompatibility protocol?
- Moreover, it would be important to have a figure showing the titration amounts on cell viability. I can only assume that the results the authors presented would be a 100% extract for their 85% cell viability result (page 7 line 150).

Comments about nerve injury terminology

- In terms of the nerve injury, I dislike the term the authors use 'acquired peripheral nerve injury'. I think it is ok to use it once at the start but it really is a compression injury and should be called out as such throughout the manuscript.
- My first glance and supplementary figure 19 depicting the nerve injury made me think this was a stretch or traction type injury. The authors should retake those images so as to not stretch the nerve out. Use background material (see any clinical peripheral nerve injury paper) to isolate the nerve and provide some contrast in the image. This will also help bring out the compression site. I understand the intent is to show the compression site in the figure but a dotted line around the zone of compression may also be helpful.

Comments about in vitro experiments

On page 8, the in vitro experiment uses what the authors call an “Electric Field”. Historically in the stimulation literature when a paper refers to electric fields it’s often a DC field.

- The manuscript is not clear what kind of field is being generated and I assumed this was DC as this is often used in vitro to provide a ‘guiding field’ for neurite extension (this is a well-known effect). I was surprised to look at supplementary figure 21 and see a standard function generator picture with a 20 Khz sine wave label. This should be reported in the manuscript instead of electrical field intensity.
- Can the authors describe why 20 Khz? This is significantly different from the stimulation nerve injury literature and should be included in the manuscript.
- Can the authors confirm that the 1.3V/mm was simply the amplitude of the function generator divided by the size of the contacting magnesium electrode?
- How did the authors determine the treatment schedule rationally to be 5d from only 72h of in vitro experimentation?
- I may have missed this but I did not see in the methods a description of the in vitro stimulation setup and output parameters, IPSC’s stimulated for 5 min only or longer? I see a caption in figure 3b but nothing in the manuscript text.
- How were the plates contacting the cells to provide stimulation if the figure (supp fig 21) says an insulating layer was used to prevent contact with the culture medium?

Comments about in vivo experiments

- It was not mentioned how long these studies are. Are nerve assessments performed immediately after 5d of stimulation treatment? This is shown as day 8 in figure 3a but is not described in the methods or the manuscript text.
- The authors mention constant pressure was applied to the nerve, was this measured? Was pressure standardized to create a consistent injury?
- How big was the zone of injury, how big were the surgical forceps?
- In the methods section (page 21, line 439), the authors should state that the electrode was placed with the anode and cathode around the crush site as shown in figure 3a. Hopefully this is a correct interpretation.
- There is no description in the methods about the walking track analysis performed (supp fig 28). Were mice trained on walking track prior to the test? Also, the image is somewhat faded and can use more contrast to bring out the paw prints.

- For figure 4a, the histomorphometry images of the wild type can be better. The axons look squished and not reflective of normal uninjured axons. In fact, the electron microscopy image of the ESE treated axons looks better than wild type except for the contrast in the image.

- The authors demonstrate dramatic improvements with stimulation therapy. Can the authors cite other papers using high frequency application of stimulation following nerve compression? It would be interesting to compare.

- There is no discussion on why the authors believe this paradigm is effective or an explanation on their dramatic results. Simply stating that it worked without providing context on why is doing readers a disservice.

Overall this paper is novel and requires some work to get to a level ready for publication especially on the selection of stimulation parameters as little detail is found in this manuscript.

Our Responses to the Comments from Reviewers

Title: An on-demand bioresorbable neurostimulator

Manuscript number: NCOMMS-23-20524-T

We appreciate the reviewers' valuable comments on our manuscript.

Reviewer # 1

In this work, the authors present an implantable ultrasonic transducer based on the principle of triboelectricity. The paper's results are very interesting. The multifunctional usage of ultrasound is the major innovation point of this paper. In this study, ultrasound was used not only as a medium for energy delivery, but also as a trigger for *in vivo* degradation. In addition, the authors performed a significant amount of biomedical experiments, which demonstrated the application prospects of implantable triboelectric energy conversion technology.

However, there are still several comments that need to be addressed by the authors. These contents mainly include the structural design of the triboelectric device, the balance between the functionality and safety of ultrasonic power, and the biocompatibility of the device etc. The specific comments are as follows.

Comment 1:

The performance comparison between the current device and the commercially-available nerve stimulators is missing. Could the authors use Bland-Altman analysis or other methods to perform such kind of comparison?

Response 1:

Authors appreciate the reviewer's insightful comment. Although our device demonstrated significant efficacy in nerve condition recovery in terms of recovery state and time to achieve effect, direct comparison with other studies is challenging. This is due to the different small animal models, signal waveforms, and target nerves employed in each study. Furthermore, to the best of our knowledge, this work represents the first attempt to explore the clinical benefits of electrical stimulation implemented by a 20 kHz output TENG. Currently, there are few commercial products or prototypes that we can compare with our device. Nevertheless, we believe that the series of *in vitro* and *in vivo* analyses presented in this work effectively demonstrate the potential of our device as a new class of implantable medical devices.

Comment 2:

The reviewer is a little bit confused on the sentence "The locally intensified acoustic pressure in the porous structure of the PHBV membrane triggers its mechanical disintegration" in Fig. S13. Could the authors provide more explanation?

Response 2:

Authors are thankful for reviewer's comment. The transmission characteristics of ultrasound waves are governed by the acoustic impedance of the medium, which can be deduced from its mechanical

properties. When ultrasound waves impinge upon an interface with another medium, the mismatch in acoustic impedance between the two distinct medium leads to the reflection of the ultrasound waves. According to the findings detailed in Supplementary Note 1, it was observed that approximately 99.94% of the acoustic wave experienced reflection at the interface of the PHBV and air, providing the localization of acoustic pressure within the pores of the PHBV membrane. To alleviate this localized acoustic pressure, mechanical disintegration is promoted in the vicinity of the PHBV porous structure. Notably, the PHBV within the inter-pore spaces is susceptible to acoustic pressure-induced mechanical stress, thereby facilitating material disintegration. We now updated this description in our revised supplementary information.

“According to the findings detailed in Supplementary Note 1, it was observed that approximately 99.94 % of the acoustic wave experienced reflection at the interface of the PHBV and air, providing the localization of acoustic pressure within the pores of the PHBV membrane. To alleviate this localized acoustic pressure, mechanical disintegration is promoted in the vicinity of the PHBV porous structure. Notably, the PHBV within the inter-pore spaces is susceptible to acoustic pressure-induced mechanical stress, thereby facilitating material disintegration.”

Comment 3:

If the unit of Fig. S13 is dB, the maximum and minimum values of acoustic pressure are 231.450 dB and 231.440 dB respectively, and the corresponding absolute pressures are 7.4736 MPa and 7.4650 MPa respectively. The difference between them is less than 0.12%. The reason why such a small acoustic pressure difference can lead to mechanical disintegration needs more detailed discussion.

Response 3:

Authors are thankful for the reviewer to address an important part to be discussed. Our proposed mechanical disintegration is mainly attributed to applying high-intensity ultrasound (HIU, 3.0 W cm^{-2}). Thus, we observed that the increase in acoustic pressure by $\sim 15.56 \text{ dB}$ (from 215.89 dB to 231.45 dB) results in the HIU-driven mechanical disintegration. In addition, we highlighted that mechanical disintegration occurs around the porous structure of PHBV due to the locally intensified acoustic pressure upon 3.0 W cm^{-2} of incident ultrasound intensity. As mentioned by the reviewer, although only a small difference of less than 0.12 % takes place between the pore and PHBV, our *in vitro* and *in vivo* demonstrations confirm that these small variations in acoustic pressure can lead to significant difference in mechanical behavior as time accumulates.

Comment 4:

Fig. 3b-c are obtained from the in-vitro cell experiments, is it the same for Fig. 3d-e? Considering that Fig. 3a is an in-vivo characterization model, the in vitro or in vivo methods used in subsequent experiments need to be explained in the manuscript.

Response 4:

Authors are thankful for the reviewer's comment. Fig. 3b and 3c describe the *in vitro* demonstration on the effect of electrical stimulation (sinusoidal waveform, 20 kHz) using IPSC-driven motor neuron. By observing a significant increase in axon length of electric field-stimulated motor neuron (5 minutes daily over a period of 3 days), it enabled establishing the *in vivo* experiment timeline as

presented in Fig. 3a. The NCS results, recorded nerve conduction velocities and action potentials displayed in Fig. 3d and 3e, were measured using the proposed *in vivo* method shown in Figure 3a. As the reviewer requested, the caption of Fig. 3 has been modified to prevent confusion in the revised manuscript.

“b,c, In vitro cell experiments were performed to validate the conditions of electrical stimulation (e.g., amplitude, treatment duration) required for neuroregeneration. (b) Optical microscopic images of iPSC-driven motor neurons. (c) Axon length plots along the culturing time.”

Comment 5:

The authors should briefly discuss the effect of the output voltage amplitude of the neurostimulator on electrotherapy for treating peripheral neuropathies. Different ultrasound power will affect the output voltage of ACT-TENG, how can the authors determine that the curative effect of 0.5 W/cm² the most suitable value?

Response 5:

Authors are thankful for addressing an important part to be discussed. We measured the ultrasound-driven triboelectric voltage upon different intensity of incident ultrasound. As shown below, there is a tendency for the voltage output generated from the ACT-TENG to increase as the ultrasound intensity increases. However, the ultrasound intensity for energy generation was determined by 0.5 W·cm⁻² based on the following points.

First, the *in vitro* cell experiments show that the axon length is saturated at the 1.3 V·mm⁻¹ (Supplementary Fig. 22). Considering that the area of compression injury lesion is approximately 2 mm, it was expected that a potential of 2.6 V would be sufficient. Besides, we confirmed that the implanted ACT-TENG can generate an adequate voltage output upon applying 0.5 W·cm⁻² of ultrasound intensity (Supplementary Fig. 24).

Next, when applying probe intensities greater than 1.5 W·cm⁻², we observed mechanical disintegration of the device, accelerating its transient processes (Supplementary Fig. 29). However, when applying probe intensities of 1.0 W·cm⁻² or lower, we found that even with prolonged ultrasound application, the transient process of the devices was not triggered. Thus, we concluded that the low probe intensities (lower than 1.0 W·cm⁻²) are advantageous for the long-term device operation. We now updated the data in our revised supplementary information.

“Supplementary Figure 30. Ultrasound-driven triboelectric output performances of ACT-TENG at various ultrasound intensity (W cm⁻²).”

Comment 6:

Supplementary Fig. 8 shows the working principle of the TENG used. The TENG appears to be a TENG in single-electrode mode. In fact, previous studies have shown that the single-electrode mode TENG is not the highest in terms of energy conversion efficiency. The charge transfer efficiency is significantly limited for this mode. The authors should add some descriptions and rationale of the TENG structure design in the text.

Response 6:

Authors are thankful for the reviewer's considerate comment. As the reviewer mentioned, TENGs generally exhibit higher energy generation performances with their structure based on multi-electrode layers. However, due to significant difference in acoustic impedance, ultrasound is predominantly reflected when it encounters metallic object in the human body (Supplementary Note), leading to significant decrease in membrane vibration and output power. Due to this limitation, ultrasound-driven TENGs were constrained to a single-electrode mode configuration. As a future work, we are focusing our efforts on developing a conductive layer that allows ultrasound transmission, aiming to construct a multi-electrode layered ultrasound-driven TENG.

Comment 7:

If this is the most suitable mode TENG, the authors should discuss the choice of structural parameters, such as the thickness of the Mg electrode, the distance between the electrode and the triboelectric layer. In particular, theoretically speaking, different distances between the electrode and the triboelectric layer will result in different TENG outputs. Then, if there is a certain most suitable stimulation voltage, can it be adjusted by changing the structural parameters while keeping the ultrasound power constant?

Response 7:

Authors appreciate valuable feedback from the reviewer's comment in terms of structural parameters and output performance. Our previous work demonstrated that the application of incident ultrasound results in the vibration of a suspended triboelectric layer with displacements in the range of several hundred micrometers (μm) (*Science* **365**, 491-494 (2019)). In this work, we also confirmed through FEM simulations that shows similar displacements ranging from tens to hundreds of μm (Supplementary Fig. 9). We further confirmed that our freestanding PHBV/PEG:ChCl membrane, placed on top of an Mg electrode in our ACT-TENG, exhibits electrical outputs with a sinusoidal waveform, indicating sufficient contact-separation motions through ultrasound-induced vibration.

As mentioned by the reviewer, triboelectric output performances are significantly influenced by various structural parameters of the devices. The following research conducted the theoretical and experimental investigation on triboelectric output performances depending on the thickness of the spacer and electrodes (*Matter* **5**, 4315-4331 (2022)). However, it is important to note that this study employed an ultrasound source with the frequency of 28 kHz. Considering that the ultrasound frequency and the materials selection of the devices can significantly affect the ultrasound-driven triboelectric effect, further in-depth research is required to address this aspect.

Comment 8:

One of the conditions for this device to work is that the hermeticity of the device remains good. The author should provide comments on the hermeticity of this device.

Response 8:

The authors greatly appreciate the insightful comment regarding the durability of the device. Previous research has demonstrated that many polyhydroxyalkanoates (PHAs), including PHBV, exhibit surface erosion behavior under physiological conditions (*Adv. Mater.* **32**, 1907138 (2020); *Polym. Degrad. Stab.* **167**, 102-113 (2019)). This behavior arises from the low water permeability and low hydrolysis reaction rate of PHBV, which distinguishes it from the majority of bioresorbable polymers that undergo bulk degradation. These unique characteristics have significant implications for the hermeticity of the device, and they form the foundation for our selection of PHBV as the material for the encapsulation layer. Primarily, the low water permeability of PHBV prevents the early infiltration or leakage of liquid physiological media into the device. Secondly, the surface degradation behavior of PHBV serves to mitigate unpredictable damage or severe mechanical degradation of the encapsulation layer. The gradual degradation from the surface over an extended period ensures a prolonged lifetime of the encapsulation layer, spanning years. We now updated this description in our revised manuscript and supplementary information.

“We utilized PHBV membranes due to their low dissolution rate (about 10 % weight loss in 450 days), allowing more practical manipulation of the device lifetime (see more descriptions about transient mechanism of PHBV membranes in Supplementary Note 2)²¹.”

“Supplementary Note 2. Macroscopic degradation mechanism of PHBV

Previous studies have provided evidence of surface erosion behavior in various polyhydroxyalkanoates (PHAs), including PHBV, when exposed to physiological conditions^{7,8}. This distinctive behavior is attributed to the inherent properties of PHBV, such as its low water permeability and hydrolysis reaction rate, making it apart from most bioresorbable polymers that undergo bulk erosion. These unique characteristics hold significant implications for ensuring the hermeticity of the device, thus influencing our choice of PHBV as the material for encapsulation layer.

First, the low water permeability of PHBV effectively prevents premature infiltration or leakage of liquid physiological media into the device. Moreover, the surface erosion behavior of PHBV serves as a protective mechanism, mitigating the risk of unexpected damage or severe mechanical disintegration of the encapsulation layer. By degrading gradually from the surface over an extended period, the PHBV ensures the stable device operation.”

*“7. Choi, S. Y., Cho, I. J., Lee, Y., Kim, Y.-J., Kim, K.-J. & Lee, S. Y. Microbial polyhydroxyalkanoates and nonnatural polyesters. *Adv. Mater.* 32, 1907138 (2020).*

*8. Salomez, M., George, M., Fabre, P., Touchaleaume, F., Cesar, G., Lajarrige, A. & Gastaldi, E. A comparative study of degradation mechanisms of PHBV and PBSA under laboratory-scale composting conditions. *Polym. Degrad. Stab.* 167, 102-113 (2019).”*

Comment 9:

There is no complex power management circuit in this paper. In this case, we believe that the intensity of electrical stimulation is affected by the intensity of ultrasound. The power density of 0.5 W cm^{-2} is used in the normal electrical stimulation, but the power density of degradation is 6 times of that. Is it possible that the intensity of electrical stimulation required for treatment needs to be large enough (considering the severity of the disease), but the corresponding ultrasound intensity at this time will already cause the device to start to disintegrate? Please explain this part.

Response 9:

We are thankful for addressing a thoughtful comment. As the reviewer mentioned, in commercially available neurostimulators, the intensity of electrical stimulation is adjusted based on the severity of the disease. However, a bioresorbable circuit is required to regulate the electrical stimulation intensity in bioresorbable neurostimulators. Yet, it remains highly challenging to fabricate bioresorbable material-based diodes and capacitors. In current state, to control the electrical stimulation intensity, we design bioresorbable triboelectric materials with high surface charge density, capable of generating high triboelectric output even at low ultrasound intensities. As a future work, we aim to integrate bioresorbable circuits and develop a technology that allows the application of electrical stimulation based on the severity of the disease.

Comment 10:

If the above situation does not exist, how to determine the optimal ultrasonic power intensity? What this paper proposes is an innovative device with a platform nature. Is it necessary to add a circuit that adjusts the intensity of electrical stimulation to adapt to different treatment intensities and treatment frequencies? (Avoid the degradation area that exists by directly adjusting the ultrasound intensity)

Response 10:

The authors are thankful for reviewer's considerate comment. We agree with the reviewer's point of view, and it would be innovative to introduce circuit systems into implantable TENGs, thereby enabling control over electrical stimulation (e.g., intensity, and frequency). However, as mentioned in the previous response, constructing circuits which can provide reliable rectification and modulation using bioresorbable materials is still challenging. In this work, we have developed a bioresorbable TENG with ultrasound-mediated device lifespan. Currently, we are in the process of developing a bioresorbable circuit that can regulate the energy generated by the body-implanted TENG.

Comment 11:

The reviewer still has concerns about the biocompatibility of the device. On the one hand, whether the concentration of Mg will cause toxicity to the human body; on the other hand, whether the intensity of ultrasound will affect the health of the human body. The authors should provide comments on the biocompatibility of such devices.

Response 11:

The authors are thankful for the reviewer's valuable comment regarding the biocompatibility of our device. Magnesium (Mg) is a well-known biocompatible and bioresorbable metal element (*J. Orthop. Traumatol.* **17**, 63-73 (2016); *J. Magnes. Alloy.* **10**, 627-669 (2022)). Its ion form (Mg^{2+})

ranks as the fourth most abundant cation in the human body and relatively large amount can be metabolized. $\text{Mg}(\text{OH})_2$, their hydrolysis product exhibits good biocompatibility as it can be absorbed by macrophages and safely excreted through urine without disrupting physiological activity (*Biomater.* **112**, 287-302 (2017); *Adv. Healthc. Mater.* **8**, 1901030 (2019); *Adv. Healthc. Mater.* **10**, 2002236 (2021)). With its secured biocompatibility, the authors believe that Mg is a rational choice as an electrode material for our bioresorbable neurostimulator.

We have adhered to the standards set by the International Electrotechnical Commission (IEC) for ultrasound-driven therapy equipment (IEC standard 60601-2-5) in our experiments. This standard imposes an upper limit of effective intensity, which is the ratio of acoustic output power to effective radiating area, at $3.0 \text{ W}\cdot\text{cm}^{-2}$. To ensure human safety, we have used ultrasound intensities not exceeding this limit. Our investigations have also revealed no instances of inflammation, heat-induced tissue damage, or hemolysis following the HIU-triggering event (Fig. 2c). Additionally, our findings from histological studies suggest that the HIU-triggering event and accelerated degradation of the device does not cause significant inflammation, heat-induced tissue damage, or hemolysis, further validating its biosafety (Fig. 2c). We now updated this description in our revised manuscript.

“The ultrasound intensity was set based on the standards outlined by the International Electrotechnical Commission (IEC) for ultrasound-driven therapy equipment (IEC standard 60601-2-5). This standard establishes a maximum threshold for the effective intensity, defined as the ratio of acoustic output power to effective radiating area, which is set at 3.0 W cm^{-2} .”

Comment 12:

HIU (more than 3.0 W cm^{-2}) triggers the in vivo disintegration of the neurostimulator, providing on-demand device elimination according to the clinical needs and situation. However, is intensity over 3.0 W cm^{-2} safe for animals or people?

Response 12:

We appreciate the reviewer’s thoughtful comment. As we mentioned in the previous response, we have followed the International Electrotechnical Commission (IEC) standards. Our investigations also showed no signs of inflammation, tissue damage, or hemolysis upon the high-intensity ultrasound triggering event. Histological studies confirmed the biosafety of the system, indicating no significant adverse effects on tissues.

Comment 13:

A more detailed introduction part will help readers grasp the technical basis of this article. The author may consider supplementing relevant literature on ultrasound based TENG and nerve electrical stimulation therapy, and compare the advantages and disadvantages of RF-based electrotherapy and ultrasound-based electrotherapy.

Response 13:

The authors would like to express our gratefulness to get the reviewer’s insightful comment. Remarkable achievements have been made in transient electronics that utilize RF-coupled energy transmission, which extends beyond neurostimulation to encompass applications as diverse as cardiac pacing and diagnostic devices. However, RF-coupled energy transmission has limitations in terms of the distance between the transmitting coil (Tx) and the receiving coil (Rx), requiring a

proximity of 5 cm or less for efficient energy transfer. Misalignment between the two coils significantly reduces the energy transmitting efficiency, which remains a challenge to overcome (*Science* **376**, 1006-1012 (2022); *Adv. Mater.* **33**, 2103974 (2021)). In addition, RF-based neurostimulators frequently utilize biodegradable diodes and various circuit elements to modulate the frequency from MHz signals (13.56 MHz). This incorporation of multiple components leads to a complex materials configuration for the device. The inherent complexity makes it challenging to accurately predict the device's lifetime, as the constituent materials exhibit diverse degradation performances.

This work represents the first application of ultrasound-driven TENG technology to neuroregeneration, employing a simple structure constructed solely of PHBV-based polymers and a Mg electrode. Demonstrated through *in vitro* and *in vivo* experiments, we verified the capability of the TENG-driven neurostimulator to undergo disintegration at a desired time using medically available ultrasound. We believe that our proposed ultrasound-triggered transient mechanism, which enables the device removal through biosafety-secured approaches, holds the potential for significant advancements in the field of transient electroceutical devices. We have diligently revised the manuscript in accordance with the comment provided.

“However, the practical application of conventional bioresorbable electronics for clinical purposes has been hindered by the inherent challenge of predicting device lifespan, which arises from the diverse transient characteristics exhibited by their constituent materials.”

Comment 14:

In the *in vitro* experiment part, PBS solution was used to verify the insolubility of the device itself and the degradation performance under ultrasonic triggering. But the Supplementary Fig. 10 shows that the experimental setup is under deionized water environment, not PBS. Can this deionized water environment be replaced with a PBS solution?

Response 14:

We appreciate the reviewer's considerate comment. The acoustic impedance, a critical parameter determining acoustic energy transfer within a medium, exhibits comparable values between PBS solution and deionized water ($1.52 \times 10^6 \text{ kg m}^{-2} \text{ s}^{-1}$ for PBS and $1.50 \times 10^6 \text{ kg m}^{-2} \text{ s}^{-1}$ for deionized water) (*Ocean Eng.* **103**, 160-170 (2015); *Ultrasound Med. Biol.* **40**, 410-421 (2014)). Therefore, deionized water was employed in the *in vitro* experiment to assess the energy generation characteristics of our ACT-TENG, which is consistent with the methodology used in our previous study (*Science* **365**, 491-494 (2019)).

Comment 15:

The unit of Fig. S13 is not marked.

Response 15:

Thank you for correcting our mistake. As the reviewer pointed out, the units are not presented for the FEM simulation results displayed in the Supplementary Fig. 13. We now updated the units in the Figure as shown below.

Reviewer # 2

The article proposes an on-demand bioresorbable neurostimulator based on triboelectric nanogenerator. An ultrasound source enables electrical stimulation and rapid device elimination. However, the evaluation of important performance is absent, the underlying mechanism of high-frequency electrical stimulation on nerve regeneration is not clear, and the histology studies demonstrating the therapeutic effects are not sufficiently convincing. These critical issues greatly diminish the quality of the work and cannot warrant publication on *Nature Communications*.

Comment 1:

The Mg electrode adopted in the work is quite thin, with a thickness of only 100 nm. This renders it susceptible to rapid degradation when exposed to physiological solutions, which can be accelerated further by even low-intensity ultrasound. However, the authors did not provide the evaluation of the operational lifetime of the device in aqueous environments. The authors should provide evidence that the device can perform over the long term to offer adequate therapeutic support. In Fig. 1, the operational window is shown up to only 120 minutes; however, this time frame does not suffice for effective nerve regeneration.

Response 1:

Authors respect the reviewer's comment. Our PHBV encapsulation layer, which exhibits a unique transient mechanism of surface erosion, allowed for the device protected from biofluids. As a result, we experimentally confirmed that our ACT-TENG with 100 nm-thick Mg electrodes showed stable energy generation performances. In addition, we demonstrated the low transient rate of PHBV and PHBV/PEG:ChCl membranes in the PBS solution (pH 7.4) at 75 °C without any triggering events (Supplementary Fig. 16). It is noteworthy that these results have shown that the membranes are quite stable in physiological settings, and they displayed no significant damage after 21 days even with accelerated transient rate at 75 °C.

To substantiate the long lifespan and robustness of the device in providing therapeutic benefits, we conducted additional demonstrations. We measured the output performances of ACT-TENG when submerged in PBS (pH 7.4, 37 °C) for 7 days, subjecting it to 0.5 W·cm⁻² ultrasound. The outcome was encouraging in that the device maintained a stable output voltage of approximately 7.8 V throughout the testing period, which we believe is adequate for effective nerve regeneration.

“Supplementary Figure 31. *The ACT-TENG was immersed in PBS (pH 7.4, 37 °C) for 7 days. a* Ultrasound-driven triboelectric output performances depending on the days the device was immersed in PBS buffer solution. The voltage output remained similar during the experiment period. *b* Summation plots of Supplementary Figure 31a.”

Comment 2:

The discussion regarding the underlying mechanism of the work appears to be weak. Typically, low-frequency electrical stimulation (below 500 Hz) is employed for neural activation, whereas high-frequency stimulation is utilized for conduction block to alleviate pain (*Sci. Adv.* **8**, eabp9169, 2022). Given that high-frequency stimulation (~20 kHz) sustains depolarization and helps to suppress action potentials, it is unclear how this technique can promote nerve regeneration which needs activation.

Response 2:

We would like to express our respect to the reviewer’s comment. In recent years, there has been a growing interest in stimulation techniques that employ supraphysiological frequencies in the kilohertz range due to their unique and diverse advantages. Their clinical application is on the rise and the range of effects is spanning from subthreshold effects to suprathreshold effects, desynchronization, and of course, conduction block is one of them. The important point is that the fundamental mechanisms driving these effects have yet to be comprehensively understood, as highlighted in a recent paper (*Brain Stimul.*, **14**, 513-530 (2021)).

To the best of our knowledge, this work is first study in exploring the impact of kilohertz-frequency stimulation on nerve regeneration. We postulate that the observed effects might be sequential results by an increased calcium influx, which is known to be instrumental in axon growth via several potential pathways (*J. Neurosci.* **30**, 3175-3183 (2010)). The pathways might include electroporation, the modulation of intracellular signaling cascades, and the activation of voltage-gated calcium channels.

Additionally, our research conducted *in vitro* and *in vivo* analyses employing well-established preclinical experimental protocols. We believe that the clinical advantages can be substantiated through substantial evaluations such as nerve conduction studies (NCS), histopathological assessments, and statistical analyses. Thus, we anticipate that this work will stimulate further investigations into the underlying mechanisms in molecular biology, engineering, and clinical applications to maximize the benefits.

Comment 3:

What is the mechanism behind the acceleration of PHBV degradation with high-intensity ultrasound? Does the polymer undergo hydrolysis or disintegration? Moreover, what are the benefits of triggered degradation? The accelerated degradation could result in the intense release of acidic monomers and metallic ions, causing toxic effects.

Response 3:

The authors are thankful for addressing an important part to be discussed. Transient devices have attracted significant interest within the field of medical applications due to their suitability for short-term treatment. These devices undergo a self-absorption process, eliminating the need for extraction. Advances in material science have led to the development of on-demand transient implantable medical devices. These devices are based on stimuli-responsive polymers or are supported by drug delivery systems. Their degradation can be triggered at the intended time to provide additional clinical options and to ensure improved lifetime control (*Matter* **3**, 1031-1052 (2020)).

While most transient systems passively decay *in vivo* over time, this new class of implantable medical devices (IMDs) is designed to disintegrate quickly upon applying the corresponding stimulus. This design ensures the stable function of the devices and mitigates the potential risk of residual materials remaining in the body (*Adv. Mater.* **27**, 3783-3788 (2015); *Sci. Rep.* **9**, 18107 (2019)). Considering these developments, an increasing number of high-impact studies on triggered degradation have been reported over the past decade (*Adv. Funct. Mater.* **27**, 1606008 (2017); *RSC Adv.* **7**, 55720-55724 (2017); *Adv. Sci.* **10**, 2204801 (2023); *Matter* **3**, 1031– 1052 (2020)), and our work aligns with the research trajectory in this area.

When it comes to biosafety, as illustrated in Fig. 1 and Supplementary Fig. 13, PHBV on-demand transience involves a sequential process of mechanical disintegration initiated by a high-intensity ultrasound (HIU)-triggering event and hydrolysis acceleration due to an enlarged surface area. Furthermore, we believe that the accelerated degradation of PHBV and PHBV/PEG:ChCl membranes does not exhibit any notable toxicity, as the mechanically disintegrated membranes upon HIU showed low cytotoxicity and insignificant genotoxicity in MTT and comet assays (Fig. 2d, e). Also, Mg is the representative biocompatible and bioresorbable metal, and it is also one of the essential components of bone structure (*J. Orthop. Traumatol.* **17**, 63-73 (2016); *J. Magnes. Alloy.* **10**, 627-669 (2022)). The overall hydrolysis degradation process of Mg is $\text{Mg} + 2\text{H}_2\text{O} \rightarrow \text{Mg}(\text{OH})_2 + \text{H}_2$ and the product of Mg degradation, $\text{Mg}(\text{OH})_2$ has good biocompatibility as it could be absorbed by macrophages and safely discharged through urine without damaging physiological activity (*Biomater.* **112**, 287-302 (2017); *Adv. Healthc. Mater.* **8**, 1901030 (2019); *Adv. Healthc. Mater.* **10**, 2002236 (2021)). Due to its biocompatibility, we utilized Mg as electrode material of our bioresorbable neurostimulator. Moreover, we could not find any significant harms to the nerve due to accelerated degradation and byproducts (Fig. 2c). That being said, we acknowledge the need for more comprehensive histological and metabolic analysis, and we consider carrying out relevant investigation in future work.

Comment 4:

The assessment of the regeneration of sciatic nerves is not rigorously performed and presented. Evaluation through histological studies is highly sensitive to the location of the measurement and careful identification of the right location is critical. The histological results of remyelination in Fig. 4 are not convincing. The authors show that the remyelination of injured nerve with ESE treatment is comparable to that of the wild type after the regrowth of only 8 days, which is impossible. Refer

to previously published papers, (for example, *Nat. Med.* **24**, 1830–1836, (2018)), it takes around 8 weeks to achieve high degree of remyelination and the recovery of EMG amplitude, on dissected sciatic nerves, which is the same injured model adopted in the current work. In fact, the CMAP of the ESE treated group is much lower compared to that of the wild type group (Fig. 3d) is an indication of low degree of remyelination.

Response 4:

The authors respect the reviewer's comment. The histological studies results can be seemed unrealistic if simply compare it to previously publication. However, recovery duration varies depending on the preclinical model used. Nerve injury can be classified to neurapraxia, axonotmesis, and neurotmesis based on the severity of damage (*Brain* **66**, 231-288 (1943)). The previous work presented in Nature Medicine used mouse model of neurotmesis which involves a full transection of the axons while our work exploited mouse model possess incomplete damage by compression, called axonotmesis. While it takes months to heal complete damage without any treatment, mouse nerve with axonotmesis is known to recover itself spanning 2 to 4 weeks. If a nerve has been severed (neurotmesis), complete healing of the damage takes several months without treatment. However, unsevered and axonotmesis mouse nerves are known to repair themselves over 2 to 4 weeks. Therefore, it is not possible to compare the treatment effect after 2 weeks. Therefore, a nerve biopsy was performed on day 8, earlier than 2 weeks. In this context of pathophysiology, we believe that high degree of remyelination and regeneration event by ESE are acceptable enough.

In the case of axonotmesis caused by compression, when nerve recovery proceeds, myelin protein is recovered first. So nerve conduction speed, and sensory nerve action potential is restored. Motor and sensory nerve conduction velocities and SNAP appear nearly normal in this mouse model. However, CMAP is time consuming because it requires many muscle cells to recover. Recovery of CMAP can only be seen when many individual muscle cells have recovered after nerve damage has been repaired. From this point of view, it can be seen that in the mouse model, motor and sensory nerve conduction velocities and SNAP are almost normal, and CMAP also shows a statistically significant recovery. When it comes to the CMAP of the ESE treated group in Figure 3d, we assume that it relates to degeneration of axonal branches after the initial partial denervation of the muscle by a pruning process. In injuries where 20-30 % of the axons are damaged, collateral branching is the primary mechanism of recovery. Also, the recovery of proximal nerve occurs followed by the axonal branches started out being formed (*Hand Clin.* **29**, 317-330 (2013); *Clinical Neurophys.* **119**, 1951–1965 (2008)). Thus, the rise of CMAP degree can take more time and we believe that this is why it exhibited lower degree of restoration compared to conduction velocity or sensory nerve condition.

Comment 5:

In Fig. S4b, the labeling of hydrogen bond is incorrect. There should not be hydrogen bonding between CH₃ and O, as hydrogen is specifically referring to the intermolecular force when a hydrogen atom bonded to a strongly electronegative atom (such as F, N, O) exists in the vicinity of another electronegative atom with a lone pair of electrons.

Response 5:

Authors are thankful for the reviewer's valuable comment. Supplementary Figure 4 represents intermolecular bindings in PHBV/PEG. The chemical structure of PHBV is ascribed to the

intermolecular interaction that involves a C-H \cdots O hydrogen bond between the C=O group in one helical structure and the CH₃ group in the other helical structure, which has been described in the reference (*Macromol.* **37**, 7203-7213 (2004)). We have also updated this description and reference in our revised supplementary information.

“The chemical structure of PHBV is ascribed to the intermolecular interaction that involves a C–H \cdots O hydrogen bond between the C=O group in one helical structure and the CH₃ group in the other helical structure².”

“2. Sato, H., Murakami, R., Padermshoke, A., Hirose, F., Senda, K., Noda, I., Ozaki, Y. Infrared spectroscopy studies of CH \cdots O hydrogen bondings and thermal behavior of biodegradable poly(hydroxyalkanoate). *Macromol.* **37**, 7203-7213 (2004).”

Comment 6:

The statistical analysis is confusing in Fig. 3d, e. There is obviously significant difference between the wild type and ESE-treated groups, but the significance is not labeled.

Response 6:

We are thankful for the reviewer’s comment that will improve the readability of our manuscript. We now added the significance between the wild type and ESE-treated groups in our revised manuscript.

Figure 3. Experimental settings for *in vivo* electrotherapy and the recorded NCS results for acquired peripheral nerve injury model and C22 mouse (hereditary peripheral neuropathy, CMT1A) model.

Reviewer # 3

This is an interesting paper describing a dissolvable stimulator used to deliver electrical stimulation to peripheral nerves. The authors have shown promising results in this paper. While my area of expertise lies in peripheral nerve stimulation and injury, I do have some comments about other aspects of the paper before I detail my comments on the stimulation and nerve injury itself.

Comment 1:

With respect to accelerating the dissolution of the stimulator, why do the authors believe that no long term residues exist once their stimulator has undergone the dissolving triggering events?

Response 1:

The authors appreciate the reviewer's thoughtful comment. Compared to the passive operation system where the materials solely rely on hydrolytic degradation, our proposed HIU-triggered transient mechanism accelerates the dissolution rate of the materials. We have demonstrated that the enlarged surface area, mechanically disintegrated upon applying HIU, contributed to our active operation system with more vigorous hydrolytic reactions. Thus, we anticipate that our approach will have a significantly lower chance of prolonged residues in the body, thereby reducing potential for negative health consequences in a great extent.

Comment 2:

Does HIU damage any tissue or mechanical deform any tissue?

Response 2:

We are thankful for the reviewer's considerate comment. We have followed the International Electrotechnical Commission (IEC) standard (IEC 60601-2-5), a regulation for ultrasound-driven therapy equipment. This standard sets an upper limit of effective intensity, defined as the ratio of acoustic output power to effective radiating area, at 3.0 W cm^{-2} . Our investigations have shown no noticeable inflammation, heat-induced tissue damage, or hemolysis following the HIU-driven triggering event (as shown in Figure 2c). We now updated this description in our revised manuscript.

"The ultrasound intensity was set based on the standards outlined by the International Electrotechnical Commission (IEC) for ultrasound-driven therapy equipment (IEC standard 60601-2-5). This standard establishes a maximum threshold for the effective intensity, defined as the ratio of acoustic output power to effective radiating area, which is set at 3.0 W cm^{-2} ."

Comment 3:

In terms of biocompatibility of the device, the authors chose to use the MTT method instead of the more widely used XTT method, why is that?

Additionally, typical biocompatibility is done to an ISO standard (10993-5) which documents MTT methods for cell culture and using L929 mouse fibroblast cells not human fibroblast cells.

Response 3:

The authors are thankful for the reviewer's helpful comment. There's no doubt that the XTT method is widely used to investigate the cytotoxicity of materials, and the ISO standard (10993-5) prescribes the use of L929. We have performed the MTT method numerous times, referring to some

references in the field of implantable biomedical electronics (*Nat. Commun.* **9**, 5349 (2018)), and consistently found high cell viability. Moreover, we noted that many previous studies have employed human fibroblast cells, and the reliability of these analyses is widely recognized due to their proximity to human tissues than the mouse fibroblast cells (L929).

In response to the reviewer's comment, we conducted additional XTT tests for further confirmation. All constituent materials demonstrated high cell viabilities over 92 %, further evidencing their high biocompatibility. We now updated the method and the data in our revised manuscript and supplementary information, respectively.

“We purchased Cell Proliferation Kit II (XTT) (Sigma-Aldrich, 11465015001) to conduct the XTT assay. Human fibroblasts (ATCC, CRL-1502) and culture medium were added upon the presence of experimental films (5 mm × 5 mm) into each well of 96-well plates (10,000 cells/well, final volume: 100 μ L). They were incubated for 3 days at 37 °C with 5 % CO₂ concentration. Before measuring the absorbance, we added 50 μ L of XTT labeling mixture per well and incubated for 18 hours at 37 °C with 5 % CO₂. Then, the absorbance of the samples was measured using a microplate (ELISA) reader to evaluate the cell viability (OD = 490 nm).”

“Supplementary Figure 32. XTT cell proliferation assay results. Since all the values are comparable to those of the control group, we confirmed cytotoxicity of the constituent polymers using XTT assay tests for 72 hours. The asterisk mark () refers to the materials that were mechanically disintegrated by HIU triggering events.”*

Comment 4:

How did the authors come up with their biocompatibility protocol?

Response 4:

We are thankful for the reviewer's comment. To assess the potential toxicity of the materials, we conducted cell viability tests using the MTT assay, genotoxicity tests using the Comet assay, and examined potential inflammation responses of the implanted device through H&E staining method.

Comment 5:

Moreover, it would be important to have a figure showing the titration amounts on cell viability. I can only assume that the results the authors presented would be a 100% extract for their 85% cell viability result (page 7 line 150).

Response 5:

Authors appreciate the reviewer's considerate comment. We evaluated the biocompatibility of constitute materials using the standardized MTT method based on the ISO standard (10993-5) and high cell viability was constantly shown in repeated experiments.

Comment 6:

In terms of the nerve injury, I dislike the term the authors use 'acquired peripheral nerve injury'. I think it is ok to use it once at the start but it really is a compression injury and should be called out as such throughout the manuscript.

Response 6:

Authors appreciate the reviewer's considerate comment. Peripheral nerve damage is broadly divided into two types as followed acquired peripheral nerve injury and hereditary peripheral nerve injury. So, we experimented with two representative cases. Therefore, the expression 'acquired peripheral nerve injury' was used. However, following the reviewer's comment, we have removed the term 'acquired' from the rest of manuscript, excluding the abstract, introduction, and figure captions.

Comment 7:

My first glance and supplementary figure 19 depicting the nerve injury made me think this was a stretch or traction type injury. The authors should retake those images so as to not stretch the nerve out. Use background material (see any clinical peripheral nerve injury paper) to isolate the nerve and provide some contrast in the image. This will also help bring out the compression site. I understand the intent is to show the compression site in the figure but a dotted line around the zone of compression may also be helpful.

Response 7:

The authors are thankful for the reviewer's helpful comment. According to the reviewer's suggestion, we have replaced the images in supplementary figure 19. We followed the method described in a recent paper (*Nat. Mater.* **22**, 895-902 (2023)). A sterilized tissue retractor was used to secure the surgical site of the sciatic nerve by retracting adjacent muscle tissues. We now updated the replaced images and this description in our supplementary information.

“Supplementary Figure 19. Magnified view of the sciatic nerve. The images represent the nerve

conditions of the wild type, compression injury model, and C22 model, respectively. We used a sterilized tissue retractor to secure the surgical site of the sciatic nerve by retracting adjacent muscle tissues.”

Comment 8:

On page 8, the in vitro experiment uses what the authors call an “Electric Field”. Historically in the stimulation literature when a paper refers to electric fields it’s often a DC field.

The manuscript is not clear what kind of field is being generated and I assumed this was DC as this is often used in vitro to provide a ‘guiding field’ for neurite extension (this is a well-known effect). I was surprised to look at supplementary figure 21 and see a standard function generator picture with a 20 kHz sine wave label. This should be reported in the manuscript instead of electrical field intensity.

Response 8:

We appreciate the reviewer’s guidance, and we admit that there might be confusion on the wordings in the context of prior research history. To address this, we clarified that we employed alternative potential field in whole demonstrations, not DC or guiding field. We also marked it is of 20 kHz sinusoidal waveforms in several parts of manuscript and figures.

“After we kept the mouse for 3 d for the recovery of the surgical wound and biological engraftments, we applied low-intensity ultrasound (0.5 W cm^{-2}) to deliver electrical impulses (20 kHz sinusoidal waveform) to the secured sciatic nerve for 5 d (5 min daily) (Supplementary Fig. 19, 20). The duration and treatment schedule of ESE were determined based on the results of in vitro cell experiments using induced pluripotent stem cell (iPSC)-driven motor neuron, in which the application of ESE resulted in remarkably improved axon growth (electric field intensity = 1.3 V mm^{-1} ; 20 kHz sinusoidal waveform) (Fig. 3b, c, Supplementary Fig. 21–23).”

“Based on the results of in vitro cell experiments, electrical impulses of 1.3 V mm^{-1} (20 kHz sinusoidal waveform, 5 min daily) are effective for accelerating the growth of iPSC-driven motor neurons. Thus, we adopted those electrical stimulation conditions in our in vivo demonstrations described in Figure 3a.”

Comment 9:

Can the authors describe why 20 kHz? This is significantly different from the stimulation nerve injury literature and should be included in the manuscript.

Response 9:

Thank the reviewer for addressing an important part to be discussed. Numerous studies in the fields of injury and electroceuticals have reported that electrical stimulation of peripheral nerves with suprphysiological frequencies can produce a range of unique effects, including desynchronization and conduction block. We have persistently been developing high-performance TENGs. Notably, our ultrasound-mediated TENGs have demonstrated their potential as a power transfer technology for various biomedical applications due to their high output power.

During the conceptualization stage of this work, we were inspired to investigate the feasibility of using ultrasound-mediated TENGs for peripheral nerve treatment. This approach is particularly attractive because it doesn't require an energy storage device, which is not

biodegradable, and because ultrasound can effectively deliver mechanical energy deep inside tissues without causing significant harm.

This study represents the first attempt to utilize ultrasound-mediated TENGs as electroceuticals for peripheral nerves. Our ACT-TENG generates a 20 kHz AC current output when a 20 kHz ultrasound wave causes the membrane to vibrate at the same frequency. Initially, we aimed to explore how the ultrasound-driven triboelectric impulses (20 kHz sinusoidal waveform) influences the physiological behavior of peripheral nerves through *in vitro* tests. The results, which demonstrated promoted regeneration of iPSC-driven motor neurons, inspired us to investigate the potential of this technology to facilitate the recovery of damaged nerves.

We believe that the series of *in vitro* and *in vivo* analyses presented in this work successfully demonstrate the potential clinical benefits of ACT-TENG. It is clear that controlling the amplitude and waveform of the signal applied to the nerve is crucial for clinical application. However, this requires the integration of biodegradable diodes and circuit elements, which remains a challenge. We are optimistic that this issue can be addressed in future research.

Comment 10:

Can the authors confirm that the 1.3V/mm was simply the amplitude of the function generator divided by the size of the contacting magnesium electrode?

Response 10:

We appreciate the reviewer's comment. When two conducting plates were placed in parallel to each other, the electric field can be simplified based on the following equation:

$$E = \frac{\Delta V}{d}$$

where ΔV refers to the potential difference between the plates and d is the distance between the plates. Thus, the electric field of 1.3 V mm⁻¹ was calculated by dividing the amplitude of the function generator (26 V) from the distance between the two Mg plates (20 mm) (please refer the experimental settings of the *in vitro* cell experiment shown in Supplementary Fig. 21).

Comment 11:

How did the authors determine the treatment schedule rationally to be 5d from only 72h of *in vitro* experimentation?

Response 11:

The authors want to express acknowledgement to the reviewer's comment. As the reviewer pointed out, there is difference in treatment duration between *in vitro* and *in vivo* analysis. As shown in *in vitro* test, there was significant enhancement in axon growth upon electrical stimulation for 3 days. We extended treatment duration to 5 days in *in vivo* test to avoid potential risk of incomplete treatment and to achieve secured demonstration by minimizing possible scarification of mouse models. Additionally, given that mouse nerves with axonotmesis injury are known to recover within 2 weeks, we anticipated that conducting *in vivo* experiments for an extended duration of more than 5 days would pose challenges in accurately comparing the therapeutic effects of electrical stimulation between control and experimental groups.

Comment 12:

I may have missed this but I did not see in the methods a description of the *in vitro* stimulation

setup and output parameters, IPSC's stimulated for 5 min only or longer? I see a caption in figure 3b but nothing in the manuscript text.

Response 12:

We are thankful for the reviewer's comment that will help improving the readability of our manuscript. The *in vitro* cell experiments were performed over a period of three days, during which electrical stimulation was administered for a duration of five minutes each day. We now added this description in our revised manuscript.

"The in vitro cell experiments were performed over a period of three days, during which electrical stimulation was administered for a duration of five minutes each day. As the lesion size of ~ 2 mm in the compression injury of the sciatic nerve, it can be inferred from the in vitro results that electrical output performance of the body-implanted device is at a sufficient level to perform the in vivo electrotherapy demonstration shown in Figure 3a (Supplementary Fig. 24)."

Comment 13:

How were the plates contacting the cells to provide stimulation if the figure (supp fig 21) says an insulating layer was used to prevent contact with the culture medium?

Response 13:

The authors are thankful for the considerate comment. Insulating layers were introduced to prevent short-circuit between metal electrodes and cell culturing media that may result in unwanted electrochemical reactions. In addition, insulated electrodes have been adopted to apply electrical impulses to observe cell behaviors (*Cancer Res.* **64**, 3288-3295 (2004)). We now added this description in our revised supplementary information.

"Insulating layers (polyimide films) were introduced to prevent short-circuit between metal electrodes and cell culturing media that may result in unwanted electrochemical reactions³."

*"3. Kirson, E.D., Gurvich, Z., Schneiderman, R., Dekel, E., Itzhaki, A., Wasserman, Y., Schatzberger, R., Palti, Y. Disruption of cancer cell replication by alternating electric fields. *Cancer Res.* 64, 3288-3295 (2004)."*

Comment 14:

It was not mentioned how long these studies are. Are nerve assessments performed immediately after 5d of stimulation treatment? This is shown as day 8 in figure 3a but is not described in the methods or the manuscript text.

Response 14:

We are thankful for the valuable comment that will improve the readability of our manuscript. On Day 8, following five days of electrical stimulation applied to the sciatic nerve, nerve conduction studies (NCS) were conducted to assess the nerve condition (Fig. 3d, e). Immediately thereafter, neural tissue was collected for the histopathological examination (Fig. 4). We now updated this experimental method in our revised manuscript.

"On Day 8, following five days of electrical stimulation applied to the sciatic nerve, nerve

conduction study (NCS) experiments were conducted to assess the nerve condition (Fig. 3d, e)."

"The neural tissue, which was collected after measuring the nerve condition through NCS experiments, was subjected to these histopathological examinations."

Comment 15:

The authors mention constant pressure was applied to the nerve, was this measured? Was pressure standardized to create a consistent injury?

Response 15:

We are thankful for addressing an important part to be discussed. To induce a compression injury with a controlled level of pressure, a specific type of surgical forceps called the HALSEY needle holder (smooth jaws, total length: 12.5 cm) was employed. The needle holder comprised a total of three stages of holder drivers, of which the second stage was utilized to apply compression on the sciatic nerve for a duration of 5 seconds. We now updated this explanation in our revised manuscript.

"To induce a compression injury with a controlled level of pressure, a specific type of surgical forceps, called the HALSEY needle holder (smooth jaws, total length: 12.5 cm) was employed. The needle holder comprised total three stages of holder drivers, of which the second stage was utilized to apply compression on the sciatic nerve for a duration of 5 seconds."

Comment 16:

How big was the zone of injury, how big were the surgical forceps?

Response 16:

Authors are thankful for the reviewer's comment. The size of the lesion where the compression injury was induced was approximately 2 mm, while the total length of the needle holder we used was 12.5 cm.

Comment 17:

In the methods section (page 21, line 439), the authors should state that the electrode was placed with the anode and cathode around the crush site as shown in figure 3a. Hopefully this is a correct interpretation.

Response 17:

We are thankful for the reviewer's comment. The bioresorbable cuff electrode of our device was placed around the compressed region of the sciatic nerve, and a pair of Mg electrodes were positioned on both sides of the compressed region to enable applying proper electrical stimulation. We now added this explanation in our revised manuscript.

"To introduce electrical stimulation events (ESE), the ACT-TENG was implanted underneath the mouse dermis, and the connected bioresorbable cuff electrode was placed around the sciatic nerve (Fig. 3a, Supplementary Fig. 18). A pair of Mg electrodes in the cuff secured the compressed lesion of the sciatic nerve for peripheral nerve injury model. Meanwhile, they wrapped around the CMT1A-diseased sciatic nerve for hereditary peripheral neuropathy model."

Comment 18:

There is no description in the methods about the walking track analysis performed (supp fig 28). Were mice trained on walking track prior to the test? Also, the image is somewhat faded and can use more contrast to bring out the paw prints.

Response 18:

The authors are thankful for the reviewer to suggest methods for better demonstration. Methods about the footprint test is described in our revised supplementary information and increased contrast of the image for better visibility.

“To conduct footprint behavior analysis of wild type and device-implanted mouse, mice are trained to walk on a white paper before the trial. The pathway was constructed with acrylic walls and all paws of mice are painted with black ink. Stride length is determined by measuring the distance between each step.”

Comment 19:

For figure 4a, the histomorphometry images of the wild type can be better. The axons look squished and not reflective of normal uninjured axons. In fact, the electron microscopy image of the ESE treated axons looks better than wild type except for the contrast in the image.

Response 19:

The authors are thankful for the reviewer’s helpful comment. We took the histomorphometry images of the wild type again and replaced the original images. We now updated the images in our revised manuscript.

Comment 20:

The authors demonstrate dramatic improvements with stimulation therapy. Can the authors cite other papers using high frequency application of stimulation following nerve compression? It would be interesting to compare.

Response 20:

We would like to express our respect to the reviewer’s comment. In recent years, there has been a growing interest in stimulation techniques that employ suprathreshold frequencies in the kilohertz range due to their unique and diverse advantages. Their clinical application is on the rise and the range of effects is spanning from subthreshold effects to suprathreshold effects, desynchronization, and of course, conduction block is one of them. The important point is that the fundamental mechanisms driving these effects have yet to be comprehensively understood, as highlighted in a recent paper (*Brain Stimul.*, **14**, 513-530 (2021)).

To the best of our knowledge, this work is first study in exploring the impact of kilohertz-frequency stimulation on nerve regeneration. We postulate that the observed effects might be sequential results by an increased calcium influx, which is known to be instrumental in axon growth

via several potential pathways (*J. Neurosci.* **30**, 3175-3183 (2010)). The pathways might include electroporation, the modulation of intracellular signaling cascades, and the activation of voltage-gated calcium channels.

Additionally, our research conducted *in vitro* and *in vivo* analyses employing well-established preclinical experimental protocols. We believe that the clinical advantages can be substantiated through substantial evaluations such as nerve conduction studies (NCS), histopathological assessments, and statistical analyses. Thus, we anticipate that this work will stimulate further investigations into the underlying mechanisms in molecular biology, engineering, and clinical applications to maximize the benefits.

Comment 21:

There is no discussion on why the authors believe this paradigm is effective or an explanation on their dramatic results. Simply stating that it worked without providing context on why is doing readers a disservice.

Response 21:

We are thankful for the reviewer's thoughtful comment. Bioresorbable electronics are of significant interest in the fields of biomedical engineering and medicine, as they eliminate the need for device extraction surgeries after fulfilling their intended clinical purposes. However, conventional devices face challenges regarding their transient mechanism, as the transient rate is solely determined by the material properties and dimensions. This can result in either premature degradation, failing to ensure the desired treatment duration, or prolonged device lifetime, leading to potential negative health consequences due to the presence of device residues. Thus, there is a growing need for protocols that can accelerate the transient rate, allowing for controlled and timely device dissolution. In this study, we have developed an on-demand neurostimulator that can actively control the lifespan of the implanted bioresorbable device using biosafe ultrasound at the desired time point. Through this work, we anticipate the personalized treatment of patients without the need for device removal surgeries, as the operation of the bioresorbable neurostimulator can be tailored according to the patient's treatment circumstances. We now updated this description in our revised manuscript.

“Bioresorbable electronics are of significant interest in the fields of biomedical engineering and medicine, as they eliminate the need for device extraction surgeries after fulfilling their intended clinical purposes. However, conventional devices face challenges regarding their transient mechanism, as the transient rate is solely determined by the material properties and dimensions. This can result in either premature degradation, failing to ensure the desired treatment duration, or prolonged device lifetime, leading to potential negative health consequences due to the presence of device residues. Thus, there is a growing need for protocols that can accelerate the transient rate, allowing for controlled and timely device dissolution. In this study, we have developed an on-demand neurostimulator that can actively control the lifespan of the implanted bioresorbable device using biosafe ultrasound at the desired time point. Through this work, we anticipate the personalized treatment of patients without the need for device removal surgeries, as the operation of the bioresorbable neurostimulator can be tailored according to the patient's treatment circumstances.”

List of changes

(1) 60^h line in page 3

The following statement was added.

However, the practical application of conventional bioresorbable electronics for clinical purposes has been hindered by the inherent challenge of predicting device lifespan, which arise from the diverse transient characteristics exhibited by their constituent materials.

(2) 101st line in page 5

The following statement was modified.

We utilized PHBV membranes due to their low dissolution rate (about 10 % weight loss in 450 days), allowing more practical manipulation of the device lifetime (see more descriptions about transient mechanism of PHBV membranes in Supplementary Note 2)²¹.

(3) 171st line in page 8

The following statement was modified.

To introduce electrical stimulation events (ESE), the ACT-TENG was implanted underneath the mouse dermis, and the connected bioresorbable cuff electrode was placed around the sciatic nerve (Fig. 3a, Supplementary Fig. 18). A pair of Mg electrodes in the cuff secured the compressed lesion of the sciatic nerve for acquired peripheral nerve injury model. Meanwhile, they wrapped around the CMT1A-diseased sciatic nerve for hereditary peripheral neuropathy model. After we kept the mouse for 3 d for the recovery of the surgical wound and biological engraftments, we applied low-intensity ultrasound (0.5 W cm^{-2}) to deliver electrical impulses (20 kHz sinusoidal waveform) to the secured sciatic nerve for 5 d (5 min daily) (Supplementary Fig. 19, 20). The duration and treatment schedule of ESE were determined based on the results of *in vitro* cell experiments using induced pluripotent stem cell (iPSC)-driven motor neuron, in which the application of ESE resulted in remarkably improved axon growth (electric field intensity = 1.3 V mm^{-1} ; 20 kHz sinusoidal waveform) (Fig. 3b, c, Supplementary Fig. 21–23). The *in vitro* cell experiments were performed over a period of three days, during which electrical stimulation was administered for a duration of five minutes each day. As the lesion size of $\sim 2 \text{ mm}$ in the compression injury of the sciatic nerve, it can be inferred from the *in vitro* results that electrical output performance of the body-implanted device is at a sufficient level to perform the *in vivo* electrotherapy demonstration shown in Figure 3a (Supplementary Fig. 24). On Day 8, following five days of electrical stimulation applied to the sciatic nerve, nerve conduction study (NCS) experiments were conducted to assess the nerve condition (Fig. 3d, e).

(4) 205^h line in page 10

The following statement was added.

The neural tissue, which was collected after measuring the nerve condition through NCS experiments, was subjected to these histopathological examinations.

(5) 246th line in page 12

The following statement was added.

Bioresorbable electronics are of significant interest in the fields of biomedical engineering and medicine, as they eliminate the need for device extraction surgeries after fulfilling their intended clinical purposes. However, conventional devices face challenges regarding their transient mechanism, as the transient rate is solely determined by the material properties and dimensions. This can result in either premature degradation, failing to ensure the desired treatment duration, or prolonged device lifetime, leading to potential negative health consequences due to the presence of device residues. Thus, there is a growing need for protocols that can accelerate the transient rate, allowing for controlled and timely device dissolution. In this study, we have developed an on-demand neurostimulator that can actively control the lifespan of the implanted bioresorbable device using biosafe ultrasound at the desired time point. Through this work, we anticipate the personalized treatment of patients without the need for device removal surgeries, as the operation of the bioresorbable neurostimulator can be tailored according to the patient's treatment circumstances.

(6) 301st line in page 14

The following statement was added.

The ultrasound intensity was set based on the standards outlined by the International Electrotechnical Commission (IEC) for ultrasound-driven therapy equipment (IEC standard 60601-2-5). This standard establishes a maximum threshold for the effective intensity, defined as the ratio of acoustic output power to effective radiating area, which is set at 3.0 W cm^{-2} .

(7) 329th line in page 16

The following statement was added.

We purchased Cell Proliferation Kit II (XTT) (Sigma-Aldrich, 11465015001) to conduct the XTT assay. Human fibroblasts (ATCC, CRL-1502) and culture medium were added upon the presence of experimental films (5 mm × 5 mm) into each well of 96-well plates (10,000 cells/well, final volume: 100 μL). They were incubated for 3 days at 37 °C with 5 % CO₂ concentration. Before measuring the absorbance, we added 50 μL of XTT labeling mixture per well and incubated for 18 hours at 37 °C with 5 % CO₂. Then, the absorbance of the samples was measured using a microplate (ELISA) reader to evaluate the cell viability (OD = 490 nm).

(8) 418th line in page 20

The following statement was added.

To induce a compression injury with a controlled level of pressure, a specific type of surgical forceps, called the HALSEY needle holder (smooth jaws, total length: 12.5 cm) was employed. The needle holder comprised total three stages of holder drivers, of which the second stage was utilized to apply compression on the sciatic nerve for a duration of 5 seconds.

(9) Figure 3 in page 31

The following figure and caption were modified.

Figure 3. **Experimental settings for *in vivo* electrotherapy and the recorded NCS results for acquired peripheral nerve injury model and C22 mouse (hereditary peripheral neuropathy, CMT1A) model.** **a** Schematic of the experimental setup for *in vivo* electrotherapy. **b,c** *In vitro* cell experiments were performed to validate the conditions of electrical stimulation (e.g., amplitude, treatment duration) required for neuroregeneration. **(b)** Optical microscopic images of iPSC-driven motor neurons. **(c)** Axon length plots along the culturing time. Based on the results of *in vitro* cell experiments, electrical impulses of 1.3 V mm⁻¹ (20 kHz sinusoidal waveform, 5 min daily) are effective for accelerating the growth of iPSC-driven motor neurons. Thus, we adopted those electrical stimulation conditions in our *in vivo* demonstrations described in Figure 3a. **d** Recorded nerve conduction velocities and action potentials after the *in vivo* electrotherapy for the acquired peripheral nerve injury mouse model (Wild type: *n* = 5; Injury model: *n* = 5; ESE-untreated: *n* = 5; ESE-treated: *n* = 4). **e** Recorded nerve conduction velocities and action potentials after the *in vivo* electrotherapy for the C22 mouse model (Wild type: *n* = 5; C22 model: *n* = 5; ESE-untreated: *n* = 5; ESE-treated: *n* = 5). *P* values are evaluated through two-sided *t* test; **P* < 0.05; ***P* < 0.01; ****P* < 0.001; *****P* < 0.0001.

(10) Figure 4 in page 32

The following figure was modified.

(0) 703rd line in page 35

The following statement was added.

Supplementary Note 2 | Macroscopic degradation mechanism of PHBV

(1) 748th line in page 40

The following statement was added.

The chemical structure of PHBV is ascribed to the intermolecular interaction that involves a C–H···O hydrogen bond between the C=O group in one helical structure and the CH₃ group in the other helical structure².

(2) Supplementary Figure 13 in page 49

The following figure and caption were modified.

According to the findings detailed in Supplementary Note 1, it was observed that approximately 99.94 % of the acoustic wave experienced reflection at the interface of the PHBV and air, providing the localization of acoustic pressure within the pores of the PHBV membrane. To alleviate this localized acoustic pressure, mechanical disintegration is promoted in the vicinity of the PHBV porous structure. Notably, the PHBV within the inter-pore spaces is susceptible to acoustic pressure-induced mechanical stress, thereby facilitating material disintegration.

(14) Supplementary Figure 19 in page 55

The following figure and caption were modified.

(0) 863rd line in page 57

The following statement was modified.

A pair of Mg plates (20 mm width, 2 mm length, 100 μm thickness) was attached to the nanopatterned cell culture dish to apply electrical impulses (20 kHz sinusoidal waveform). Insulating layers (polyimide films) were introduced to prevent short-circuit between metal electrodes and cell culturing media that may result in unwanted electrochemical reactions³.

(3) Supplementary Figure 28 in page 64

To conduct footprint behavior analysis of wild type and device-implemented mouse, mice are trained to walk on a white paper before the trial. The pathway was constructed with acrylic walls and all paws of mice are painted with black ink. Stride length is determined by measuring the distance between each step.

The following figure and caption were modified.

(4) Supplementary Figure 30 in page 66

The following figure and caption were added.

(5) Supplementary Figure 31 in page 67

The following figure and caption were added.

Supplementary Figure 30. **Ultrasound-driven triboelectric output performances of ACT-TENG at various ultrasound intensity (W cm⁻²).**

(6) Supplementary Figure 32 in page 68

The following figure and caption were added.

(7) Supplementary Note 2 in page 70

The following statement was added.

Supplementary Note 2. Macroscopic degradation mechanism of PHBV

Previous studies have provided evidence of surface erosion behavior in various polyhydroxyalkanoates (PHAs), including PHBV, when exposed to physiological conditions^{7,8}. This

distinctive behavior is attributed to the inherent properties of PHBV, such as its low water permeability and hydrolysis reaction rate, making it apart from most bioresorbable polymers that undergo bulk erosion. These unique characteristics hold significant implications for ensuring the hermeticity of the device, thus influencing our choice of PHBV as the material for encapsulation layer.

First, the low water permeability of PHBV effectively prevents premature infiltration or leakage of liquid physiological media into the device. Moreover, the surface erosion behavior of PHBV serves as a protective mechanism, mitigating the risk of unexpected damage or severe mechanical disintegration of the encapsulation layer. By degrading gradually from the surface over an extended period, the PHBV ensures the stable device operation.

(21) References in page 71

The following references were added.

2. Sato, H., Murakami, R., Padermshoke, A., Hirose, F., Senda, K., Noda, I. & Ozaki, Y. Infrared spectroscopy studies of CH \cdots O hydrogen bondings and thermal behavior of biodegradable poly(hydroxyalkanoate). *Macromol.* 37, 7203-7213 (2004).
3. Kirson, E.D., Gurvich, Z., Schneiderman, R., Dekel, E., Itzhaki, A., Wasserman, Y., Schatzberger, R. & Palti, Y. Disruption of cancer cell replication by alternating electric fields. *Cancer Res.* 64, 3288-3295 (2004).
7. Choi, S. Y., Cho, I. J., Lee, Y., Kim, Y.-J., Kim, K.-J. & Lee, S. Y. Microbial polyhydroxyalkanoates and nonnatural polyesters. *Adv. Mater.* 32, 1907138 (2020).
4. Salomez, M., George, M., Fabre, P., Touchaleaume, F., Cesar, G., Lajarrige, A. & Gastaldi, E. A comparative study of degradation mechanisms of PHBV and PBSA under laboratory-scale composting conditions. *Polym. Degrad. Stab.* 167, 102-113 (2019).

REVIEWERS' COMMENTS

Reviewer #1 (Remarks to the Author):

The reviewer has carefully read the revised version of the manuscript entitled "An on-demand bioresorbable neurostimulator

". In this revised version, the authors have amended the ambiguity and added Supplementary Note to strengthen the clarity and novelty of this manuscript. The reviewer appreciates the revision and recommend a possible publication of this article.

Reviewer #2 (Remarks to the Author):

The authors addressed most of my questions. However, it would have been clearer if the authors had indicated the changes made in response to each comment. This lack of clarity makes it difficult to determine whether the authors have revised the manuscript accordingly. Additionally, I noticed that some discussions mentioned in the response letter were not incorporated into the manuscript. For instance, it is important to include a discussion on the mechanism by which high frequency stimulation promotes nerve repair, as also suggested by reviewer 3. I also agree with the suggestion to replace the term "acquired peripheral nerve injury" with "compression injury" to avoid confusion. This change should be applied consistently throughout the manuscript, including figure legends ("acquired peripheral nerve injury" is still used in Figure 3).

Reviewer #3 (Remarks to the Author):

Thank you for addressing my comments and those of the other reviewers. While I still think there is a lack of reasoning in the paper for choosing 20 Khz for regeneration, I would suggest in future articles the authors start to investigate the mechanism of action behind the efficacy of this high frequency since I am not aware of any other literature supporting this utility. The article is acceptable with the current changes.

Our Responses to the Comments from Reviewers

Title: An on-demand bioresorbable neurostimulator

We appreciate the reviewers' valuable comments on our manuscript.

Reviewer # 1

The reviewer has carefully read the revised version of the manuscript entitled "An on-demand bioresorbable neurostimulator". In this revised version, the authors have amended the ambiguity and added Supplementary Note to strengthen the clarity and novelty of this manuscript. The reviewer appreciates the revision and recommend a possible publication of this article.

Response:

The authors are deeply appreciative of the reviewer's comments on our manuscript. The thorough evaluation undertaken by the reviewer has been instrumental in identifying and revising areas of ambiguity, thus leading to a significant enhancement in the clarity of the content. The positive recommendation from the reviewer is both encouraging and motivating, further reinforcing the author's commitment to producing high-quality research.

Reviewer # 2

The authors addressed most of my questions. However, it would have been clearer if the authors had indicated the changes made in response to each comment. This lack of clarity makes it difficult to determine whether the authors have revised the manuscript accordingly. Additionally, I noticed that some discussions mentioned in the response letter were not incorporated into the manuscript. For instance, it is important to include a discussion on the mechanism by which high frequency stimulation promotes nerve repair, as also suggested by reviewer 3. I also agree with the suggestion to replace the term "acquired peripheral nerve injury" with "compression injury" to avoid confusion. This change should be applied consistently throughout the manuscript, including figure legends ("acquired peripheral nerve injury" is still used in Figure 3).

Response:

The authors are thankful for the reviewer's valuable comments in the original revision process. We also appreciate the reviewer's thoughtful suggestions to improve the clarity of our manuscript. In response, we updated the discussion on the mechanism of the high-frequency stimulation-driven neuroregeneration in our revised manuscript as follows.

"In this study, we undertook comprehensive investigations into the therapeutic impact of kilohertz-frequency stimulation on neuroregeneration. We postulated that the observed effects may be attributed to a sequence of events involving an elevated calcium influx, a phenomenon recognized for its pivotal role in axon growth through various potential pathways²⁹. These pathways encompass potential mechanisms such as electroporation, modulation of intracellular signaling cascades, and the activation of voltage-gated calcium channels."

“29. Ghosh-Roy, A., Wu, Z., Goncharov, A., Jin, Y. & Chisholm, A. D. Calcium and cyclic AMP promote axonal regeneration in Caenorhabditis elegans and require DLK-1 kinase. J. Neurosci. 30, 3175-3183 (2010).”

In accordance with the recommendation provided by the reviewer #3, we have revised the term ‘acquired peripheral nerve injury’ to ‘compression nerve injury’ throughout our manuscript. This modification is expected to enhance the precision of our manuscript and mitigate any potential confusion. We express our appreciation for the valuable suggestion.

We highlighted all the changes made in this revision process in our revised manuscript.

Reviewer # 3

Thank you for addressing my comments and those of the other reviewers. While I still think there is a lack of reasoning in the paper for choosing 20 kHz for regeneration, I would suggest in future articles the authors start to investigate the mechanism of action behind the efficacy of this high frequency since I am not aware of any other literature supporting this utility. The article is acceptable with the current changes.

Response:

The authors express sincere gratitude for the thorough consideration of the reviewer. Despite the increasing attention and significance attributed to the therapeutic effects of neural tissue through electrical stimulation, their biological mechanism remains elusive. As advised by the reviewer, the authors aspire to undertake further investigations in our future research endeavors to elucidate the role of electrical signals in manifesting neural tissue regeneration.

List of changes

(1) 38^h line in page 2

The following statement was revised.

Furthermore, neurophysiological analyses support that our neurostimulator provides remarkable therapeutic benefits for both compression peripheral nerve injury and hereditary peripheral neuropathy.

(2) 83rd line in page 4

The following statement was revised.

We place a particular focus on its therapeutic effect in both compression peripheral nerve injury and hereditary peripheral neuropathy including Charcot–Marie–Tooth disease (CMT).

(3) 89^h line in page 5

The following statement was revised.

Although kHz frequency stimulation with sinusoidal waveform has been employed as a peripheral nerve stimulation (PNS) system by engaging ion channel dynamics of the targeted axons, its therapeutic effect on both compression nerve injury and hereditary peripheral neuropathy has remained undiscovered^{19,20}.

(4) 165^h line in page 8

The following statement was revised.

We performed in vivo experiments to investigate the therapeutic effects of the on-demand bioresorbable neurostimulator on both compression peripheral nerve injury and CMT1A mouse models.

(5) 186^h line in page 9

The following statement was revised.

In the NCS experiments for the compression peripheral nerve injury model, the ESE significantly increased both motor nerve conduction velocities (MNCVs) and sensory nerve conduction velocities (SNCVs) (Fig. 3d).

(6) 203rd line in page 10

The following statement was revised.

For the compression peripheral nerve injury model, we confirmed that the ESE-treated group represented a higher degree of myelination than the peripheral nerve injury group.

(7) 206^h line in page 10

The following statement was revised.

Note that the ESE treatment resulted in an increased number of myelinated axons as the average diameter of the axons increased, and the distribution curve shifted to the right than the compression peripheral nerve injury group.

(0) 221st line in page 11

The following statement was revised.

These statistical demonstrations support that the kHz frequency electrical stimulation results in the myelination of the neuronal axons for both compression peripheral nerve injury and hereditary peripheral neuropathy including CMT1A.

(1) 224^h line in page 11

The following statement was added.

In this study, we undertook comprehensive investigations into the therapeutic impact of kilohertz-frequency stimulation on neuroregeneration. We postulated that the observed effects may be attributed to a sequence of events involving an elevated calcium influx, a phenomenon recognized for its pivotal role in axon growth through various potential pathways²⁹. These pathways encompass potential mechanisms such as electroporation, modulation of intracellular signaling cascades, and the activation of voltage-gated calcium channels.

(2) 244th line in page 12

The following statement was revised.

The high-frequency electrical stimulation (20 kHz, 7.76 V) introduced distinct improvements in the conduction velocities and action potentials for both the compression peripheral nerve injury and hereditary CMT1A mouse models.

(3) 427^h line in page 20

The following statement was revised.

Animal preparation (the compression peripheral nerve injury mouse model)

(4) 565th line in page 26

The following reference was added.

29. Ghosh-Roy, A., Wu, Z., Goncharov, A., Jin, Y. & Chisholm, A. D. Calcium and cyclic AMP promote axonal regeneration in *Caenorhabditis elegans* and require DLK-1 kinase. *J. Neurosci.* **30**, 3175-3183 (2010).

(13) Figure 3 in page 31

The following figure and caption were revised.

Figure 3. Experimental settings for in vivo electrotherapy and the recorded NCS results for compression peripheral nerve injury model and C22 mouse (hereditary peripheral neuropathy, CMT1A) model. **a** Schematic of the experimental setup for in vivo electrotherapy. **b,c** In vitro cell experiments were performed to validate the electrical stimulation conditions (e.g., amplitude, treatment duration) required for neuroregeneration. **(b)** Optical microscopic images of iPSC-driven motor neurons. **(c)** Axon length plots along the culturing time (n = 3 for each group). Data are presented as mean values \pm SD. Based on the results of in vitro cell experiments, electrical impulses of 1.3 V mm⁻¹ (20 kHz sinusoidal waveform, 5 min daily) are effective for accelerating the growth of iPSC-driven motor neurons. Thus, we adopted those electrical stimulation conditions in our in vivo demonstrations described in Figure 3a. **d** Recorded nerve conduction velocities and action potentials after the in vivo electrotherapy for the compression peripheral nerve injury mouse model (Wild type: n = 5; Injury model: n = 5; ESE-untreated: n = 5; ESE-treated: n = 4). Data are presented as mean values \pm SD. ESE represents electrical stimulation events. **e** Recorded nerve conduction velocities and action potentials after the in vivo electrotherapy for the C22 mouse model (Wild type: n = 5; C22 model: n = 5; ESE-untreated: n = 5; ESE-treated: n = 5). Data are presented as mean values \pm SD. P values are evaluated through two-sided t test; *P < 0.05; **P < 0.01; ***P < 0.001; ****P < 0.0001.

The following figure and caption were revised.

Figure 4. **Histopathological demonstration and statistical analyses for the therapeutic effects on both compression peripheral nerve injury and CMT1A (hereditary peripheral neuropathy).** **a** Semithin sections (scale bar: 30 μm) and electron microscopy images (scale bar: 2 μm) to observe the toluidine blue-stained sciatic nerve cross-section. **b** Histogram representing the inner diameter distribution of the myelinated axons (Wild type: $n = 3$; Injury model: $n = 3$; ESE-treated: $n = 3$). Data are presented as mean values \pm SD. **c** Histopathological demonstration using semithin (scale bar: 30 μm) and electron microscopy images (scale bar: 2 μm) of the sciatic nerve cross-section. **d,e** Percentage of the myelinated axons (**d**) and the unmyelinated axons (**e**) (the myelinated axons refer to the axons with a diameter greater than 5 μm) (Wild type: $n = 3$; C22 model: $n = 3$; ESE-untreated: $n = 3$; ESE-treated: $n = 3$). Data are presented as mean values \pm SD. **f** Histogram displaying the diameter distribution (Wild type: $n = 3$; C22 model: $n = 3$; ESE-untreated: $n = 3$; ESE-treated: $n = 3$). Data are presented as mean values \pm SD. **g** Scatterplots representing the correlation of g-ratio and axon diameter (Wild type: $n = 900$; C22 model: $n = 800$; ESE-untreated: $n = 200$; ESE-treated: $n = 416$). P values are evaluated through a two-sided t test; ns = non significant; * $P < 0.05$; **** $P < 0.0001$.